# Group Fairness in Reinforcement Learning via Multi-Objective Rewards

**Jack Blandin**                                                                    *blandin1@uic.edu*
*Department of Computer Science*
*University of Illinois Chicago*

**Ian Kash**                                                                         *iankash@uic.edu*
*Department of Computer Science*
*University of Illinois Chicago*

**Reviewed on OpenReview:** *https://openreview.net/forum?id=cueEUSG7lE*

## Abstract

Recent works extend classification group fairness measures to sequential decision processes such as reinforcement learning (RL) by measuring fairness as the difference in decision-maker utility (e.g. accuracy) of each group. This approach suffers when decision-maker utility is not perfectly aligned with group utility, such as in repeat loan applications where a false positive (loan default) impacts the groups (applicants) and decision-maker (lender) by different magnitudes. Some works remedy this by measuring fairness in terms of group utility, typically referred to as their "qualification", but few works offer solutions that yield group qualification equality. Those that do are prone to violating the "no-harm" principle where one or more groups' qualifications are lowered in order to achieve equality. In this work, we characterize this problem space as having three implicit objectives: maximizing decision-maker utility, maximizing group qualification, and minimizing the difference in qualification between groups. We provide a RL policy learning technique that optimizes for these objectives directly by constructing a multi-objective reward function that encodes these objectives as distinct reward signals. Under suitable parameterizations our approach is guaranteed to respect the "no-harm" principle.

## 1 Introduction

In this work, we develop an approach for learning policies which satisfy *group fairness* definitions in reinforcement learning (RL), where an algorithm is considered fair if its results are independent of one or more protected attributes such as gender, ethnicity, or sexual-orientation. There is by now an extensive body of work on group fairness works in classification settings (Berk et al., 2018; Chouldechova, 2017; Corbett-Davies et al., 2017; Dwork et al., 2012; Hardt et al., 2016; Kusner et al., 2017; Galhotra et al., 2017). Moving beyond classification, Liu et al. (2018) initiated the study of measuring the downstream impact of one-shot fairness constraints on the individuals they aim to protect by evaluating the change in credit score in a two-step loan application model. Several works build on this by analyzing how the *qualification* of individuals changes over time as a function of various decision-based fairness constraints (D'Amour et al., 2020; Mouzannar et al., 2019; Zhang et al., 2020). Although these works study long-term qualification impact, they do not offer techniques for learning policies that obtain long-term qualification equality across protected groups. Others works attempt policy-learning techniques that result in long-term qualification equality, but suffer from one or both of the following two drawbacks:

**Drawback 1.** *The solution is prone to violating the "no-harm" principle (Martinez et al., 2020) where one or more group's qualification is lowered in order to satisfy the equality constraint, without any qualification improvement to another group.*

Wen et al. (2021) for instance, maximize discounted decision-maker utility subject to a group qualification equality constraint, and focus on repeat loan applications. Even though they distinguish decision-maker utility and group qualification as separate objectives, we show that their approach is still prone to violating the no-harm principle because there is no penalty for lowering a group's qualification in order to satisfy the constraint. Our primary goal with respect to avoiding harm is to navigate the balance between improving the qualifications of one group without unjustly compromising the qualifications of another.

**Drawback 2.** *The solution assumes that decision-maker utility is equal to group qualification, which leads to solutions that are suboptimal with respect to decision-maker utility or group qualification.*

Several works consider fairness as some measure of overall utility (Martinez et al., 2020; Diana et al., 2021; Hu & Zhang, 2022; Chi et al., 2021; Siddique et al., 2020). For example, Martinez et al. (2020) characterize group fairness as a multi-objective optimization problem where each sensitive group risk is a separate objective, and their solution minimizes the maximum error across all groups. This approach, as well as the others, assumes that the decision-maker and the groups share the exact same objective. In prison sentencing, for instance, the decision-maker (judge) and the groups (defendants) have conflicting objectives. If we are measuring group fairness, we should measure it with respect to the *group* objectives, not the decision-maker objective. In addition, practical solutions need to balance decision-maker utility with group fairness: solutions that only consider fairness will have poor decision-maker utility, and vice versa.

We seek a fair policy-learning solution that remedies the aforementioned drawbacks. Therefore, our objective is to find a policy-learning technique that improves long-term group qualification equality; does not lower one or more group qualification without improving another group's qualification; and is robust to situations when qualification and decision-maker utility are different functions. In order to ensure that qualification is improved and that qualification equality is maintained, we seek a technique that optimizes for these values directly as objectives in a multi-objective reward.

## 1.1 Related Work

**Applying one-shot constraints in sequential decision processes.** Deng et al. (2022) propose a method for applying fairness constraints at each decision point in sequential RL settings, diverging from our approach of integrating multi-objective rewards. Hu & Chen (2020) study fairness when there are multiple decisions for an individual, but measure the one-shot fairness of each decision instead of long-term fairness. These methods enforce statistical independence of decisions from sensitive attributes at every timestep, a strategy that may not align with achieving nuanced fairness due to the non-IID nature of RL, as highlighted by the analysis of Liu et al. (2018). Therefore, they are all prone to violating the "no-harm" principle.

**Characterizing the long-term fairness impact of one-shot constraints.** Several works build on Liu et al. (2018)'s two-step analysis of downstream impact of one-shot fairness constraints by analyzing how the *qualification* of individuals change over time as a function of various decision-based fairness constraints. D'Amour et al. (2020) extend this study beyond two timesteps and measure one-shot impact on long-term qualification rates through a simulated loan application RL environment. Mouzannar et al. (2019) study the impact of affirmative action on qualification rates over many timesteps. Zhang et al. (2020) measure the impact of one-shot constraints on qualification rate disparity of the policy's equilibrium. Although these works study long-term qualification impact, they do not offer techniques for learning policies that obtain long-term qualification equality across protected groups.

**Learning policies that achieve long-term fairness in sequential decision settings.** In addition to the group risk minimax approach of Martinez et al. (2020), other works attempt to learn policies that achieve long-term fairness. Hu & Zhang (2022) propose a structural causal model framework for achieving long-term fair policies for sequential decision making, formulated as a constrained optimization problem with the decision-maker utility as the objective and both long-term and short-term fairness as constraints. Chi et al. (2021) minimize reward disparity by minimizing differences in state visitation frequencies between group-specific policies. Diana et al. (2021) build on that of Martinez et al. (2020), but offer a solution that can relax the fairness constraints, thus permitting tradeoffs between decision-maker utility and fairness. Siddique et al. (2020) consider fair multi-objective MDPs and try to satisfy the generalized Gini social welfare function

(Weymark, 1981). However, each of these works suffer from Drawback 2, so their definition of "fairness" is with respect to a utility function that may not align with the groups it aims to protect.

Raab & Liu (2021) study local fairness constraints and their impact on equilibrium qualification rates. Also focusing on a repeat-loan application environment, they seek an algorithm that eliminates qualification differences in equilibrium, but assume unfairness is only due to initial population differences and that the groups are otherwise have equivalent behavior. Thus any unfairness introduced as a result of differing qualification dynamics will go unresolved.

Satija et al. (2023) study fairness in RL settings where the objective is to maximize decision-maker reward subject to a difference constraint minimization with respect to the group utility. However, because only the difference is being optimized, this technique is also prone to violating the no-harm principle.

### 1.2 Contributions

We propose a technique that optimizes for decision-maker utility, group qualification, and group qualification equality directly as a distinct objectives in a mult-objective reward. While our technique can in principle extend to a variety of environments, we focus on the RL setting. We construct a weighted sum of three distinct reward functions for decision-maker utility, qualification improvement, and qualification equality. By framing our technique as a reward function, rather than a policy intervention, any RL algorithm may learn the optimal policy since the reward adheres to the standard RL paradigm.

While on its own our objective does not guarantee the "no-harm" property, we show how to parameterize our multi-objective approach to guarantee no harm to one or more of the groups. Similar to Liu et al. (2018), D'Amour et al. (2020), Wen et al. (2021), and Zhang et al. (2020), we demonstrate our approach on a sequential loan application MDP environment, and benchmark our results against two state-of-the-art techniques. While we restrict our experimental settings to a loan application environment, we provide a model and approach that generalizes to a much broader area of applications where a decision-maker evaluates individuals based on an observed metric, and where the decision itself influences this observed metric in the future.

Our work builds on the line of research initiated by Liu et al. (2018) who first articulated how local fairness constraints do not ensure long-term fairness in sequential decision-making. Subsequent studies by Mouzannar et al. (2019) and Zhang et al. (2020) investigate when such constraints can actually promote long-term fairness, offering insights but not developing specific policy methods. Our research builds on these findings, and propose a policy-learning strategy to address this gap. Therefore, our model of the environment is intentionally similar to that of Zhang et al. (2020), Mouzannar et al. (2019), among others Wen et al. (2021); D'Amour et al. (2020).

Other policy-learning strategies like those from Wen et al. (2021), Hu & Zhang (2022), and Chi et al. (2021) seek long-term fairness, but often lower some groups' qualifications without broader benefits, which implies they violate the no-harm principle. Martinez et al. (2020) first proposed a policy-learning strategy to solve for this problem of harm, by providing a policy-learning solution that maximizes the minimum group qualification. Our method offers a more efficient alternative, demonstrating that while their approach prevents harm, ours achieves better solutions. In other words, our primary goal with respect to avoiding harm is to navigate the balance between improving the qualifications of one group without unjustly compromising the qualifications of another. Achieving fairness often requires adjusting the qualifications of the more advantaged group to aid the disadvantaged group significantly. This is the inherent challenge of fairness, since if it were always possible to improve one or more groups without consequence, achieving fairness would be trivial.

## 2 Model

We illustrate our model with a running example (also used later in our experiments) of a sequential lending scenario with a single lender and a population of loan applicants. At each timestep $t$, an *individual* (e.g. loan applicant) is sampled and the *decision-maker* (e.g. lender) makes a decision (e.g. to either approve or reject

the sampled individual's loan application).[1] We consider two groups of individuals who are distinguished by their *sensitive attribute* $z \in \mathcal{Z}$ (e.g. gender). At time $t$ we can sample an individual whose sensitive attribute $z \sim Z_t$ is governed by its time-invariant distribution $P(Z)$.[2] In addition to their protected attribute, an individual is characterized by their *qualification* $y^z \sim Y_t^z$ (e.g. loan repayment probability or credit score). As noted by Zhang et al. (2020), this model can be interpreted as either representing randomly selected individuals repeatedly going through the decision cycles, or the population-wide average when all individuals are subject to the decision cycles. So $Y^z$ could be the probability of an individual in group $z$ having qualification $y^z$ at time $t$, while also being the average qualification of group $z$ at time $t$.

As with prior work on long-term fairness, we assume individuals care about optimizing their qualification (D'Amour et al., 2020; Mouzannar et al., 2019; Zhang et al., 2020). The lender has access to the applicant's protected attribute (e.g., gender) and other non-sensitive attributes (e.g., income, occupation, marital status). The outcome of the lender's decision (e.g., whether the applicant repaid the loan, defaulted, or was rejected) affects the applicant's credit score in future timesteps. The lender also keeps track of the total population's credit scores over time.

While we restrict our Section 5 experiments to the repeat loan application studied by Zhang et al. (2020) [3], our approach generalizes to many other domains as well, including recidivism prediction and parole decisions (Dressel & Farid, 2018; Imai & Jiang, 2020), predictive policing (Ensign et al., 2018), affirmative action and the labor market (Mouzannar et al., 2019), and food inspections (Singh et al., 2022; D'Amour et al., 2020). We formally define our model as a Markov Decision Process.

**Definition 2.1** (Markov decision process). *A Markov Decision Process (MDP) is a 6-tuple $\{S, \mathcal{A}, T, R, \gamma, \mu\}$ where $S$ is a set of states; $\mathcal{A}$ is a set of actions; $T : S \times \mathcal{A} \to \Delta S$ is a mapping of state-action pairs to a distribution over new states: $T(s_t|s_{t-1}, a_{t-1})$; $R : S \times \mathcal{A} \times S \to \mathbb{R}$ is the reward function, which maps a state-action-state triplet to a real number; $\gamma \in [0,1]$ is the discount factor; and $\mu$ is the initial state probability distribution. A typical goal is to find the optimal policy $\pi^* \in \Pi$ that maximizes the expected discounted reward, where $\Pi$ is the space of policies.*

The decision-maker is represented by the *policy* $\pi$ which selects an *action* $a \sim A_t \in \mathcal{A}$ (e.g. reject/accept) based on the current state $\{z, y^0, y^1, y^\Delta, x\}$, where $z \sim Z_t$ is the protected attribute of the sampled individual at time $t$; $y^0 \sim Y_t^0$ and $y_t^1 \sim Y_t^1$ are the qualifications (e.g. credit scores) of the two groups at time $t$; $y^\Delta \sim Y_t^\Delta$ is the cumulative qualification difference between groups (used to define our notion of fairness in Section 3) at time $t$: $Y_k^\Delta = \sum_{t=0}^{t=k} Y_t^1 - \sum_{t=0}^{t=k} Y_t^0$; and $x \sim X_t$ is a the *environment* state components at time $t$, which represents all other attributes of the state excluding qualification and sensitive attributes (e.g. the lender's available cash).

The action $a$ and the individual's qualification $y^z$ inform the *outcome* $\theta \sim \Theta_t$ (e.g. repaid/defaulted) by way of the *outcome dynamics* $T^\Theta(y^z, a) := P(\theta \mid y^z, a)$. The outcome $\theta$ impacts the qualification of the individuals in the subsequent timestep, $y^{0'} \sim Y_{t+1}^0$ and $y^{1'} \sim Y_{t+1}^1$, according to the *qualification dynamics* of each group $T^{Y^0}(y^0, y^1, \theta) := P(y^{0'} \mid y^0, y^1, \theta)$ and $T^{Y^1}(y^0, y^1, \theta) := P(y^{1'} \mid y^0, y^1, \theta)$. The outcome also impacts the environment state (e.g. increase/decrease the cash available to make future loans) in the subsequent timestep $x' \sim X_{t+1}$ according to the *environment dynamics* $T^X(x, \theta) := P(x' \mid x, \theta)$. The decision-maker wants to optimize their utility (e.g. lender wants to optimize profit). We denote the *decision-maker utility function* as $R^D : \Theta \to \mathbb{R}$. We assume a finite time-horizon with $\gamma = 1$. [4] The full graphical model is depicted in Figure 1.

---

[1] The sequential loan model delineated in our study presupposes that a decision-maker must await the observed outcome of a prior decision before proceeding with a subsequent one. Nonetheless, our reward framework is not predicated on such temporal constraints; it is based on the expected sum of future rewards, which remains unaffected by the specific timing of these rewards.

[2] We use the notation $x \sim X$ to denote $x$ being a sample from (the law of) the random variable $X$.

[3] Our loan application model also shares similarities to those of D'Amour et al. (2020) and Wen et al. (2021).

[4] Although we use a value of $\gamma < 1$ in our experiments to illustrate policies considering longer-time horizons.

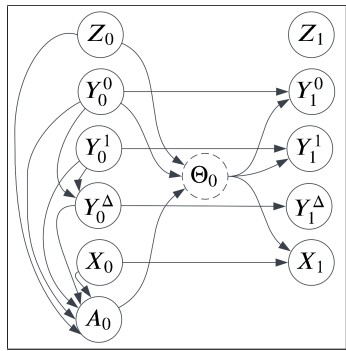

Figure 1: The graphical representation of our MDP model. Without loss of generality, we set $z \sim Z_0 = 0$ with probability 1. The left-hand side nodes represents the initial state $t = 0$ and the right-hand side nodes represents the subsequent timestep $t = 1$. The same model is repeated for all subsequent timesteps $t > 1$. The dashed circle in the middle represents the *outcome* variable $\Theta$ (e.g. repayment, default).

For brevity, we denote a state transition from $t$ to $t + 1$ after taking action $a_t$ as $\sigma_t = (s_t, a_t, s_{t+1})$. We also use $\sum_{\pi} v$ to denote the expected discounted sum of some observable quantity $v_t = v(\sigma_t) \to \mathbb{R}$ under policy $\pi$:

$$\sum_{\pi} v = \mathbb{E}_{T,\pi} \left[ \sum_{t=0}^{\infty} \gamma^t v_t \right] .$$

## 3 Multi-Objective Rewards

Here we introduce our multi-objective reward framework for learning policies that achieve fairness (i.e. qualification equality between groups) without suffering the Section 1 drawbacks. In order to balance qualification improvement and qualification equality without significant decision-maker utility degradation, we propose a technique that optimizes for these values directly. We propose a multi-objective reward function that is a weighted sum of three distinct reward functions for decision-maker utility, qualification improvement, and qualification equality. By framing our technique as a reward function, rather than a policy intervention, we can leverage any planning or RL algorithm to learn the optimal policy. Furthermore, by including terms for decision-maker utility, qualification, and qualification equality directly, this approach organically seeks efficient tradeoffs across the three objectives, and without the need for domain-specific adjustments, such as those by Raab & Liu (2021) and Hu & Chen (2020).

This multi-objective reward is defined as a weighted sum:

$$R(\lambda, \sigma_t) = \lambda^D R^D(\sigma_t) + \lambda^Q R^Q(\sigma_t) + \lambda^F R^F(\sigma_t) . \tag{1}$$

where $\lambda = (\lambda^D, \lambda^Q, \lambda^F)$ is a 3-tuple of non-negative preference weights for the decision-maker utility, qualification, and qualification equality, respectively.[5] $R^D(\sigma_t)$ denotes the decision-maker utility function as defined in Section 2, and therefore represents the reward component for decision-maker utility. $R^Q(\sigma_t)$ is the sum of group qualifications at time $t + 1$:

$$R^Q(\sigma_t) = Y_{t+1}^0 + Y_{t+1}^1 . \tag{2}$$

$R^F(\sigma_t)$ is computed as the cumulative difference in qualification averages between groups, which is equivalent to computing the difference in $Y_t^\Delta$ and $Y_{t+1}^\Delta$:

$$R^F(\sigma_t) = 1 + |Y_t^\Delta| - |Y_{t+1}^\Delta| . \tag{3}$$

---

[5]While our optimization is unconstrained, these weights also have a natural interpretation as Lagrange multipliers. Also, while our approach involves changing the *rewards*, it can also be viewed as an instance of the common RL practice of adding auxiliary losses.

We include the 1 term to ensure that the value is greater than or equal to zero. Equation (3) implies that higher values of $R^F$ correspond to smaller historical differences in group qualifications, and are therefore "more fair".

Thus Equation (1) is a weighted sum of reward contributions characterizing the extent to which the transition $\sigma_t$ contributes towards the decision-maker utility, group qualification, and group qualification equality. We denote Equation (1) parameterized by $\lambda^D = \mathtt{i}, \lambda^Q = \mathtt{j}, \lambda^F = \mathtt{k}$ as $R^{\mathtt{ijk}}$ and its optimal policy as

$$\pi_{\mathtt{ijk}} = \underset{\pi \in \Pi}{\mathrm{argmax}}\ \mathbb{E}\Big[\sum_{t=0}^{\infty} \gamma^t R^{\mathtt{ijk}}(s_t, \pi(s_t), s_{t+1})\Big]\ . \tag{4}$$

We explicitly structure Equation (1) as a sum over the three implicit objectives, which avoids the drawbacks of Section 1 by combining competing objectives into the same overall objective. Specifically, including components for overall group qualification $R^Q$ and group qualification equality $R^F$ naturally avoids harm, since harm would decrease $R^Q$. We show in Section 4 that certain combinations of weights guarantee versions of no harm. Additionally, combining the separate fairness objective $R^F$ and decision-maker objective $R^D$ into the same objective naturally finds solutions that are both fair and beneficial to the decision-maker while not conflating decision maker utility and qualification. Furthermore, the weights $\lambda^D$, $\lambda^Q$, and $\lambda^F$ allow practitioners to tune the reward function to better reflect any domain-specific aspects.

## 4 Analysis

Following convention, e.g. (Feldman et al., 2015; Kamishima et al., 2012), we enable practitioners to specify their fairness preferences with a tuning parameter, which we refer to as $\lambda$. In our approach, the practitioner specifies $\lambda$ as part of the reward function definition, which conveniently reduces the multi-objective reward to a single objective reward. The reduced single objective reward is a linear weighted sum over the specified $\lambda$ preference weights, and thus yields a single deterministic stationary policy (Roijers et al., 2013). It is for this reason that our approach fits nicely with the traditional RL paradigm since we end up with a single objective reward.

Because preference weights are selected, as opposed to computed, it is important that the practitioner understand the range of outcomes for each parameter specification. We believe that one particularly important situation is taking an existing policy that was not fairness-aware and adjusting the optimization to include fairness. One particular concern when doing so is that, unless care is taken, optimizing for fairness can harm (i.e. reduce the qualification of) the very groups that are intended to be protected (Martinez et al., 2020). Therefore, we characterize the space of outcomes for various $\lambda$ permutations, and do so based on their harm guarantees for one or more of the disadvantaged and advantaged groups. *In particular, we provide $\lambda$ configurations that guarantee no harm to the disadvantaged group, the advantaged group, or both groups.* We start with formal definitions for *harm* and for *(dis)advantaged groups.*

**Definition 4.1** (Harm). *A policy $\pi'$ does **harm** to group $Z = z$ if the difference in the group's expected discounted qualification value $Y^z$ produced by $\pi'$ relative to that produced by the decision-maker utility-optimal policy $\pi_{100}$ is less than zero:*

$$\sum_{\pi'} Y^z - \sum_{\pi_{100}} Y^z < 0\ . \tag{5}$$

*Similarly, we say that a policy $\pi'$ does **no harm** to group $Z = z$ if $\sum_{\pi'} Y^z - \sum_{\pi_{100}} Y^z \geq 0$, and **benefits** group $Z = z$ if $\sum_{\pi'} Y^z - \sum_{\pi_{100}} Y^z > 0$.*

**Definition 4.2** (Advantaged, Disadvantaged, Natural Fairness). *Group $Z = z$ is **advantaged** if $\sum_{\pi_{100}} Y^z > \sum_{\pi_{100}} Y^{1-z}$, and is **disadvantaged** if $\sum_{\pi_{100}} Y^z < \sum_{\pi_{100}} Y^{1-z}$. If $\sum_{\pi_{100}} Y^z = \sum_{\pi_{100}} Y^{1-z}$, then the environment has **natural fairness** and neither group is considered advantaged or disadvantaged.*

Without loss of generality, we let $Z = 0$ denote the group that is disadvantaged under the decision-maker utility-optimal policy.

### 4.1 No-harm Properties

Here we provide the harm guarantees for various configurations of the preference weights $\lambda$. A practitioner who wishes to implement fairness can select their choice of $\lambda$ based on the appropriate level of harm guarantees for their problem domain as well as their goals in trading off between utility and fairness.

**Theorem 4.1.** *When natural fairness does not exist, no harm is done to the disadvantaged group if $\lambda^Q = \lambda^F$.*

*Proof Sketch.* Consider a multi-objective policy $\pi_\lambda$ that is optimal under Equation (1) for some $\lambda$ configuration. In order for $\pi_\lambda$ to do harm, two conditions must hold. First, either $\sum_{\pi_\lambda} Y^0 - \sum_{\pi_{100}} Y^0 < 0$ or $\sum_{\pi_\lambda} Y^1 - \sum_{\pi_{100}} Y^1 < 0$. Second, $\pi_\lambda$ must deviate from the decision-maker utility-optimal policy $\pi_{100}$, which means that

$$\lambda^Q(\sum_{\pi_\lambda} R^Q - \sum_{\pi_{100}} R^Q) + \lambda^F(\sum_{\pi_\lambda} R^F - \sum_{\pi_{100}} R^F) \geq \lambda^D(\sum_{\pi_{100}} R^D - \sum_{\pi_\lambda} R^D) \,. \tag{6}$$

After substituting the $R^Q$ and $R^F$ values with their $Y^z$ values in Equations (2) and (3), and some simple algebra, we can show that the change in qualification of the disadvantaged group is upper-bounded by:

$$\sum_{\pi_{100}} Y^0 - \sum_{\pi_\lambda} Y^0 \leq \frac{1}{\lambda^Q + \lambda^F}\left[(\lambda^Q - \lambda^F)(\sum_{\pi_\lambda} Y^1 - \sum_{\pi_{100}} Y^1) - \lambda^D(\sum_{\pi_{100}} R^D - \sum_{\pi_\lambda} R^D)\right] \tag{7}$$

Because no policy can have higher $\sum R^D$ than $\pi_{100}$, we know that $\sum_{\pi_{100}} R^D - \sum_{\pi_\lambda} R^D \geq 0$. Therefore, the entire right-hand-side of Equation (7) will be non-positive, which implies no harm to the disadvantaged group, if

$$(\lambda^Q - \lambda^F)(\sum_{\pi_\lambda} Y^1 - \sum_{\pi_{100}} Y^1) \leq 0 \,. \tag{8}$$

This is guaranteed to occur when $\lambda^Q = \lambda^F$. $\qquad\square$

When $\lambda^Q = \lambda^F$, any increase to the advantaged group's qualification induces a positive change in $R^Q$, but an equal and opposite change in $R^F$ due to the increased qualification inequality between the two groups. This is a useful property because it fixes a lower bound on group qualification at $\sum_{\pi_{100}} Y^0$, which is the disadvantaged group's qualification under the decision-maker utility-optimal policy. If we do not consider decision-maker utility at all and set $\lambda^D = 0$, then setting $\lambda^Q = \lambda^F$ is equivalent to maximizing the minimum group qualification:

**Corollary 4.1.** *$\pi_{011}$ is equivalent to maximizing the minimum group qualification.*

This is interesting because maximizing the minimum group qualification is one way of translating the harm-avoiding fairness technique proposed by Martinez et al. (2020). to our setting [6].

**Theorem 4.2.** *At least one group is not harmed if $\lambda^Q \geq \lambda^F$.*

*Proof Sketch.* Starting with Equation (6) and using the same substitutions as the previous proof, we can show that the change in qualification of the advantaged group is non-negative if

$$(\lambda^Q > \lambda^F) \wedge (\sum_{\pi_\lambda} Y^0 - \sum_{\pi_{100}} Y^0 \leq 0) \,. \tag{9}$$

Combining Equation (8) and (9), we get Theorem 4.2. $\qquad\square$

**Theorem 4.3.** *When natural fairness exists, neither group is harmed if $\lambda^Q \leq \lambda^F$.*

If we are designing a solution for a system where we know natural fairness exists, then Theorem 4.3 becomes useful because we can encourage fairness with $\lambda^Q \leq \lambda^F$ without fear of harming either group.

We defer the full proofs for Theorems 4.1 and 4.2, and the entire proof for Theorem 4.3, to Appendix A. These proofs provide a wider range of no-harm conditions, of which we believe the ones presented here are the most

---

[6]The other way to translate their approach is to maximize the minimum decision-maker utility generated by each group. However, this translation does not make sense to use this as a fairness measure when group qualification is known to be different from decision-maker utility.

natural. Also, although the aforementioned definitions only consider harm relative to the decision-maker utility-optimal policy $\pi_{100}$, we show in Appendix B that our approach can be extended to arbitrary policies as well. However, we focus the rest of our approach with respect to $\pi_{100}$ because our most prominent concern is to avoid unnecessarily causing harm to the very groups we aim to protect when introducing fairness. While our theorems assume that we find the exact optimal policy, we discuss in Appendix C how their guarantees degrade gracefully with approximate solutions. While our approach focuses on the two-group setting, it can be extended to the more general case, though the theoretical properties become more challenging to analyze. We include a discussion in Appendix D.

## 4.2 Qualification and Fairness Tradeoffs

Here we analyze how optimal policies under the Equation (1) multi-objective reward will make tradeoffs between qualification and fairness. We do so by understanding how various parameterizations of $\lambda^Q$ and $\lambda^F$ will reward all possible pairs of group qualification outcomes $(Y_\lambda^0, Y_\lambda^1)$ for a given pair of decision-maker utility-optimal group qualification outcomes $(Y_{100}^0, Y_{100}^1)$. Figure 2 shows the "desirability" of each possible $(Y_\lambda^0, Y_\lambda^1)$ outcome for three different pairs of decision-maker utility-optimal group qualification outcomes $(Y_{100}^0, Y_{100}^1)$ and three different $\lambda^Q/\lambda^F$ ratios. The "desirability" is the net reward improvement for the qualification and fairness components relative to the decision-maker utility-optimal policy, holding the decision-maker utility reward component $R^D$ constant:

$$\Delta R^{QF} = \lambda^Q \Delta R^Q + \lambda^F \Delta R^F .$$

Darker contours indicate more attractive (higher $\Delta R^{QF}$) values. The colorless areas thus indicate where $\Delta R^{QF} \le 0$, which means they are not feasible solutions under $\pi_\lambda$ since the decision-maker utility-optimal policy will be preferred over them. The bottom three plots correspond to natural fairness settings.

When $\lambda^Q = \lambda^F$ (center column) the feasible solutions with a given $\Delta R^{QF}$ form a right-angle frontier (i.e. the set of $\Delta R^{QF} = 0$). For instance, consider the bottom-center plot. This plot represents an environment where a policy optimizing for decision-maker utility only would result in $\Sigma Y^0 = 0.5$ and $\Sigma Y^1 = 0.5$, which is shown as the blue dot at those same coordinates, and represents a scenario where natural fairness exists. Each "pixel" of the plot represents a different pair of $Y^0$, $Y^1$ outcomes, and the color represents the magnitude of the reward for those outcomes, where the reward is defined by $\lambda^Q = 1, \lambda^F = 1$, and $\lambda^D = d$ for some constant $d$. In this instance, the only way an outcome would be considered "better" according to the reward function is if both $Y^0$ and $Y^1$ increased, as represented by the darker contours only occurring up and to the right of the blue dot. This is an illustration of Theorem 4.3, which states that no group is harmed if $\lambda^Q \le \lambda^F$ and natural fairness exists.

We can also see that the disadvantaged group's qualification cannot be lowered in any of the center column plots, illustrating Theorem 4.1 that $\lambda^F = \lambda^Q$ guarantees no harm to the disadvantaged group. Furthermore, notice in each of the three center column plots that the contour does not change along any given horizontal or vertical axis on either side of the diagonal. This is because of the equal and opposite reward change in $R^Q$ and $R^F$ induced by increases in the advantaged group's qualification when $\lambda^Q = \lambda^F$. Since decision-maker utility is not considered in Figure 2, the center three plots are visualizations of $R^{dxx}$ for any constants $d$ and $x$, whose optimal policy is equivalent to maximizing the minimum group qualification.

When $\lambda^Q > \lambda^F$ (left column), the feasible solutions form a convex set with the decision-maker utility-optimal policy as a point along the frontier. This illustrates the Theorem 4.2 claim that at least one group's qualification is not harmed if $\lambda^Q \ge \lambda^F$. However, it is important to note that harm can be done to the disadvantaged group by greatly improving the advantaged group, and vice versa.

When $\lambda^Q < \lambda^F$ (right column), both the advantaged group and disadvantaged group can be harmed, unless natural fairness exists (bottom right). If natural fairness exists, then neither group will be harmed, as per Theorem 4.3.

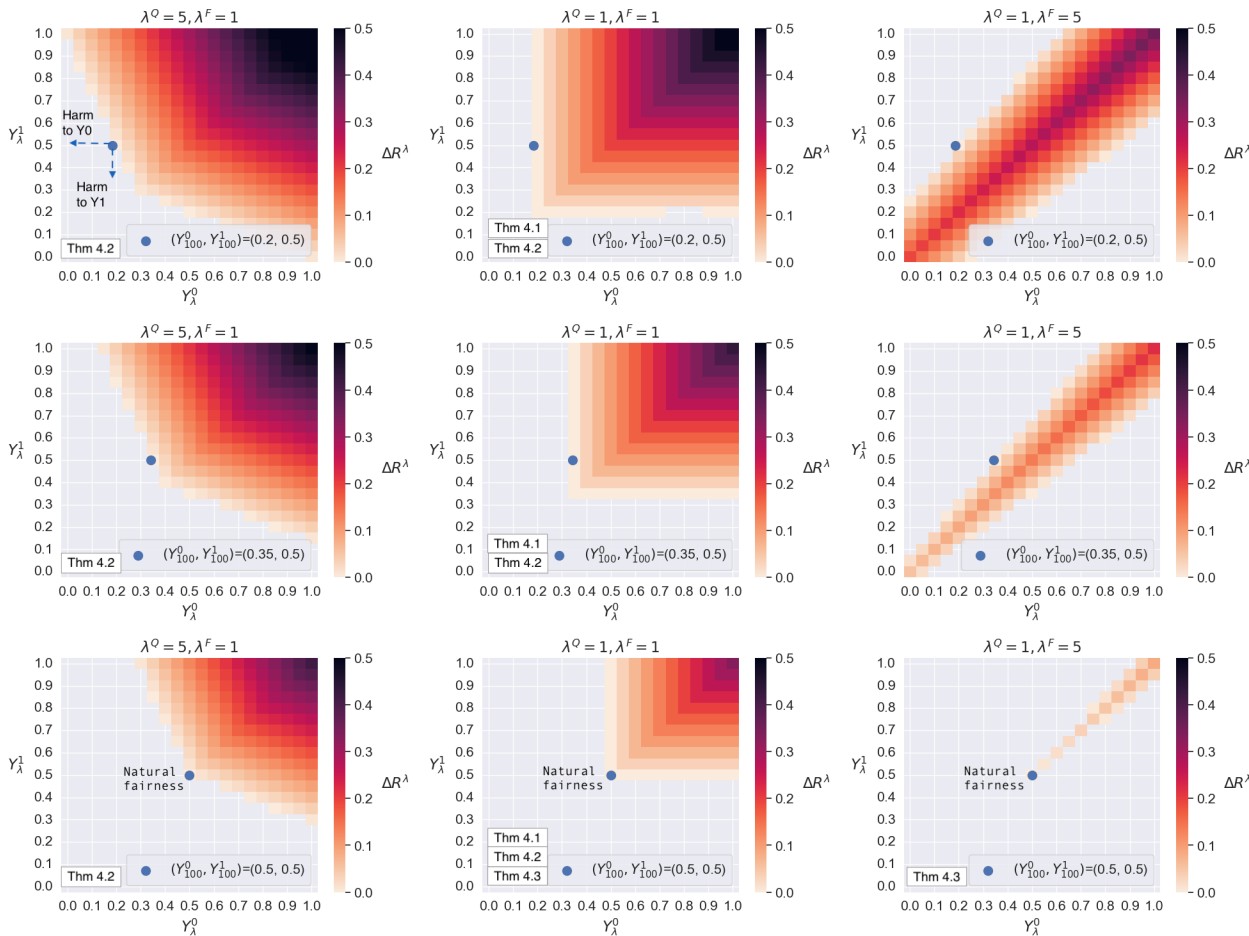

Figure 2: Plots showing the "desirability" (i.e. $\Delta R^{QF}$) of each possible $(Y_\lambda^0, Y_\lambda^1)$ outcome for three different decision-maker utility-optimal $Y$ values and three different $\lambda^Q/\lambda^F$ ratios. The blue dots indicate the $Y$ values for the decision-maker utility-optimal policy, which will by definition have $\Delta R^{QF} = 0$. Darker contours indicate more attractive (higher $\Delta R^{QF}$) values. The white areas thus indicate where $\Delta R^{QF} \leq 0$, which means they are not possible solutions under $\pi_\lambda$ since the decision-maker utility-optimal policy will be preferred over them. The lower-left corner of each plot specifies which, if any, of Theorems 4.1-4.3 are illustrated in that plot. The plots assume $\Delta R^D = 0$ so that decision-maker utility is still optimal and any $\Delta R^\lambda$ originates exclusively from $\Delta R^Q$ and $\Delta R^F$.

## 4.3 Choosing Weights in Practice

Notice that Theorems 4.1, 4.2, and 4.3 all contain the case when $\lambda^Q = \lambda^F$. Under this parameterization, we are guaranteed several beneficial properties. Specifically, we are guaranteed (i) no harm to the disadvantaged group, (ii) that any decrease in qualification for the advantaged group will result in a qualification increase for the disadvantaged group, and (iii) no harm to both groups when natural fairness exists. Therefore, we see choosing $\pi_{1xx}$ for some $x > 0$ as a natural parameterization of our framework. And since $\pi_{100}$ corresponds to when fairness is not considered at all, a practitioner can scale $x$ progressively higher from $0 < x < \infty$ until their desired balance of fairness and decision-maker utility is met, in the same manner that many fair classification algorithms have a hyperparameter controlling this same tradeoff (Feldman et al., 2015; Kamishima et al., 2012). In Section 5 we show through experiments that this parameterization consistently provides well-balanced solutions.

While there are many benefits to our single scalar reward formulation, in some scenarios it may be beneficial to decompose the reward. For instance, a practitioner may need to iterate through different weight values to achieve a satisfactory balance between decision-maker utility and fairness objectives, particularly in complex MDP scenarios. This iterative process, while manageable for simpler models, could indeed become computationally expensive as complexity rises. To address this, a practitioner could alternatively use a Multi-Objective Markov Decision Process (MOMDP) framework, which is easy to do with our reward structure since the single scalar reward can simply be segmented into three distinct rewards. By decomposing our single scalar reward into three distinct rewards, one can leverage MOMDP techniques such as the Conditioned Network algorithm Abels et al. (2019) to efficiently learn multiple policies in parallel. So a practitioner may easily transition to a MOMDP reward if they deem it necessary, without having to redefine the reward function altogether.

## 5 Experiment

Here we provide three experiments, where our goals are to demonstrate (i) Theorems 4.1-4.3 in action, (ii) how our reward encourages fairness (beyond just avoiding harm), and (iii) how our reward can lead to better policies than benchmarks. Our experiments, mirroring repeat loan applications with transition probabilities similar to those in Zhang et al. (2020) and Wen et al. (2021), ensure alignment with relevant literature.

### 5.1 Setup

We consider instantiations of the MDP defined in Section 2 and model a two-step loan application environment. [7] For clarity, we keep the problem domain consistent as a two-step loan application environment. However, in order to demonstrate a variety of settings, we provide three different sets of qualification dynamics which results in three different MDP instantiations.

Building on the two-step loan application MDP outlined in Section 2, the lender is represented by a policy $\pi$ which can, at each of the two time steps $t$, either approve the applicant's loan ($a = 1$) or reject it ($a = 0$), where $a \sim A_t$. The sampled applicant at time $t$ has a binary protected attribute $z \sim Z_t \in \{0, 1\}$ and a credit score $y^z \sim Y_t^z \in \{0, 1, 2\}$. An applicant's credit score defines their repayment probability. An applicant's credit score and protected attribute determine the probability of the applicant's credit score increasing or decreasing in the event of a repaid, defaulted, or rejected loan. Therefore, an applicant's protected attribute does not influence repayment probability, but does influence qualification dynamics.

While only the applicant from one protected group requests a loan per timestep, the decision-maker also observes the credit score of the applicant from the other group $y^{1-z} \sim Y_t^{1-z}$, and so is always able to observe the current credit scores of both groups. Our model follows the same structure as that of Zhang et al. (2020), and can similarly be interpreted as two individuals (one from each group) repeatedly applying for loans, or as a stylized model of population-level decisions and credit-score evolution. [8] We set $\gamma = 1/2$ given the short length of our episodes. We consider the applicant's credit score as their *qualification* attribute, and so we aim to improve overall applicant credit scores as well as ensure credit score equality across groups.

To make the model more realistic, the lender is limited by the number of loans it can give, which we do by limiting the lender's available cash, which corresponds to the environment state $x \sim X_t \in \{0, 1, 2\}$. The decision-maker utility $R^D$ is to maximize *profit*, which is the change in cash. If the applicant is approved for a loan and repays it, the lender receives a positive profit, and their available cash increases $x' = \min\{x + 1, 2\}$ with probability 1. Similarly, if the applicant is approved for a loan and defaults, the lender receives a negative profit, and their available cash decreases $x' = x - 1$ with probability 1. When $x = 0$, the lender is out of cash, and they may not give out loans. The lender having finite cash is significant since it constrains their decisions so that loans cannot simply be granted without consequences. So the lender may need to strategically maintain a sufficient amount of cash in order to enable later loan approvals. In addition to optimizing for their own financial gain, the lender cares about fairness, and so they also want to minimize the difference in group credit scores over time.

---

[7]The supporting code for these experiments is available at `https://github.com/jackblandin/research/`.
[8]We assume the former interpretation in our discussions.

Next we describe the initial state, outcome dynamics, and the qualification dynamics.

**Initial State**    There is equal probability of sampling an applicant from either group. The lender's available cash is $x = 1$ with probability 1. Each group's initial credit score has an equal probability of being any one of the three possible credit scores. These values are specified in Table 1 for clarity.

**Outcome Dynamics**    If the lender approves a loan then the applicant will either repay the loan ($\theta = \texttt{Repay}$) or default ($\theta = \texttt{Default}$). These probabilities are the same for all three experiments, and are included in Table 1. We only include the $\texttt{Repay}$ probabilities since the $\texttt{Default}$ are just the complements. The lender can also reject a loan or have insufficient cash to approve a loan, which result in the $\theta = \texttt{Rejected}$ or $\theta = \texttt{NoCash}$ outcomes, respectively. The $\texttt{Rejected}$ and $\texttt{NoCash}$ outcomes occur with probability 1 given the relevant action or state.

**Qualification Dynamics**    The base qualification dynamics are defined as follows. When an applicant from group $z$ repays a loan ($\theta = \texttt{Repay}$), the credit score of group $z$ increases by one in the subsequent timestep, unless they are already at the maximum credit score ($y^z = 2$), in which case they remain at 2. When an applicant from group $z$ defaults on a loan ($\theta = \texttt{Default}$), their credit score decreases by one in the subsequent timestep, unless they are already at the minimum credit score ($Y^z = 0$), in which case they remain at 0. In Section 5.1.1 we describe and motivate three specific scenarios which differ from these base dynamics in specific cases.

### 5.1.1 Qualification Dynamic Scenarios

In order to demonstrate the robustness of our approach, we evaluate three different scenarios of our experiment, where each scenario refers to a different qualification dynamic configuration, and do so in ways similar to Zhang et al. (2020) and Mouzannar et al. (2019). We show how just by changing the qualification dynamics, existing fairness policy-learning benchmarks fall short.

**Scenario 1: Conflict of interest.**    The first scenario represents a conflict of interest between the decision-maker (lender) and the disadvantaged applicant group ($z = 0$). It is an instantiation of the *demographic-variant transition* scenario studied by Zhang et al. (2020) where the disadvantaged group needs to exhaust its financial resources more so than the advantaged group in order to repay its loans, which can harm its credit score even if the loan is repaid. Therefore, a good solution will need to balance decision-maker utility with qualification equality, but not simply by lowering the advantaged group's qualification.

**Scenario 2: Natural fairness.**    This scenario represents a paradigm as in business loan borrowing where advantaged applicants depend on consistent access to loans, and will suffer financially if they are repeatedly rejected. Here, the advantaged group has a higher probability of increasing its credit score upon successful repayment, but also has a higher probability of its credit score decreasing when rejected. By construction, in this scenario the decision-maker utility optimal policy $\pi_{100}$ induces equal qualification for both groups, and therefore represents a naturally fair setting.

**Scenario 3: Credit decay.**    Scenario 3 reflects a common consumer borrowing paradigm where credit scores can change even if there is no borrowing activity. The advantaged group benefits more from this relative to the disadvantaged group. Also, the advantaged group has greater average benefits after loan repayment than their disadvantaged counterparts.

Table 1 shows the deviations from the baseline qualification dynamics in each experiment scenario.

### 5.2 Benchmark Techniques

In addition to comparing our multi-objective approach with the decision-maker utility-optimal policy ($\pi_{100}$), we also compare it against two baseline algorithms from recent fairness literature.

The first baseline policy-learning technique, $\texttt{EqOp}$, optimizes the decision-maker utility-only reward subject to a constraint that qualified applicants from each group have a difference in average credit scores less than

| | Parameter(s) | Scen 1 | Scen 2 | Scen 3 |
|---|---|---|---|---|
| Init. State | $P(z = i) \; \forall i \in \{0, 1\}$ | .50 | .50 | .50 |
| | $P(x = 1)$ | 1.0 | 1.0 | 1.0 |
| | $P(y^z = i \mid z) \; \forall i \in \{0, 1, 2\}$ | .333 | .333 | .333 |
| Outc. Dyn. | $P(\theta = \texttt{Repay} \mid z = 0, y^0 = 0, a = 1)$ | .25 | .25 | .25 |
| | $P(\theta = \texttt{Repay} \mid z = 1, y^1 = 0, a = 1)$ | .25 | .25 | .25 |
| | $P(\theta = \texttt{Repay} \mid z = 0, y^0 = 1, a = 1)$ | .67 | .67 | .67 |
| | $P(\theta = \texttt{Repay} \mid z = 1, y^1 = 1, a = 1)$ | .67 | .67 | .67 |
| | $P(\theta = \texttt{Repay} \mid z = 0, y^0 = 2, a = 1)$ | .75 | .75 | .75 |
| | $P(\theta = \texttt{Repay} \mid z = 1, y^1 = 2, a = 1)$ | .75 | .75 | .75 |
| Qual. Dyn. | $P(y^{0'} = y^0 + 1 \mid z = 0, \theta = i) \; \forall i \in \{\texttt{Rejected,NoCash}\}$ | — | — | .05 |
| | $P(y^{1'} = y^1 + 1 \mid z = 1, \theta = i) \; \forall i \in \{\texttt{Rejected,NoCash}\}$ | — | — | .25 |
| | $P(y^{0'} = y^0 - 1 \mid z = 0, \theta = i) \; \forall i \in \{\texttt{Rejected,NoCash}\}$ | — | — | .30 |
| | $P(y^{0'} = y^0 + 1 \mid z = 0, \theta = \texttt{Repay})$ | .00 | .60 | .60 |
| | $P(y^{1'} = y^1 + 1 \mid z = 1, \theta = \texttt{Repay})$ | — | — | .90 |
| | $P(y^{0'} = y^0 - 1 \mid z = 0, \theta = \texttt{Repay})$ | 1.0 | — | .20 |
| | $P(y^{1'} = y^1 - 1 \mid z = 1, \theta = \texttt{Repay})$ | — | — | .05 |

Table 1: Probability specifications for the initial state, transition dynamics, and qualification dynamics for experiment Scenarios 1-3. For the qualification dynamics, we only specify deviations from the base behavior (Section 5.1) A "—" indicates that scenario does not modify the base behavior in that case. Lines with no deviations in any scenario are omitted.

some allowable margin $\epsilon$:

$$\pi_{\texttt{EqOp}} = \operatorname*{argmax}_{\pi \in \Pi} R^D \quad \text{s.t.} \quad \left| \sum_t^{\infty} \gamma^t [Y_t^1 \mid Y_{t=0}^1 \geq \alpha] - \sum_t^{\infty} \gamma^t [Y_t^0 \mid Y_{t=0}^0 \geq \alpha] \right| < \epsilon \tag{10}$$

where $\epsilon = .05$, and an individual is considered qualified if they are more likely to repay a loan than to default, which corresponds to $\alpha = 1$ in our experiments. $\texttt{EqOp}$ tries to balance fairness with decision-maker utility by only trying to be fair to a subset of applicants. This approach is an adaptation of an Equal Opportunity constraint (Hardt et al., 2016) proposed by Wen et al. (2021). Our choices of $\epsilon$ and $\alpha$ follow the logic used by Wen et al. (2021).

The second baseline algorithm is $\texttt{MMQ}$ which reflects the minimax risk technique proposed by Martinez et al. (2020). Also, we modify the approach to maximize the minimum value since we aim to maximize qualification, rather than minimize risk:

$$\pi_{\texttt{MMQ}} = \operatorname*{argmax}_{\pi \in \Pi} [\min(\Sigma Y^0, \Sigma Y^1)] \; . \tag{11}$$

This technique tries to avoid harm by maximizing the disadvantaged (minimum) group's qualification. This is the same approach that we referenced in Section 4 that is equivalent to using our multi-objective approach with $\lambda^D = 0$ and $\lambda^Q = \lambda^F > 0$.

The MDP is simple enough that we can compute optimal policies for the multi-objective policies and $\texttt{EqOp}$ with linear programming. Leveraging Corollary 4.1, $\texttt{MMQ}$ is computed by solving for $\pi_{011}$.

### 5.3 Results

Next we provide the results for the aforementioned scenarios.

#### 5.3.1 Scenario 1

The results of Scenario 1 are shown in Figure 3. As illustrated in Figure 3a, `EqOp` harms both the disadvantaged group and advantaged group in its attempt to be fair. `EqOp` does this because it ends up making poor decisions for the advantaged group in order to drop their credit scores down to the level of the disadvantaged group. In other words, it violates the no-harm principle in order to obtain equality. On the other hand, each of the multi-objective approaches with $\lambda^Q \geq \lambda^F$ improves the qualification of at least one group, demonstrating Theorem 4.2, with several of them improving both groups. Interestingly, as shown in Figure 3b, `EqOp` actually does not improve overall qualification equality, relative to the decision-maker utility-optimal policy. This occurs because `EqOp` explicitly requires equal qualification for "qualified" individuals ($Y^z > 1$), and ignores any qualification discrepancy for "unqualified" individuals. This is consistent with the results of Zhang et al. (2020) who also observe that Equal Opportunity exacerbates qualification inequality under the demographic-variant transition scenario. Furthermore, while `MMQ` does well to improve the disadvantaged group's qualification, it makes a significant reduction in decision-maker utility in order to do so. In contrast, the multi-objective approaches with $\lambda^D$ greater than zero ($\pi_{511}, \pi_{111}, \pi_{151}, \pi_{115}$) all improve the advantaged and disadvantaged groups' qualification values with a smaller reduction in decision-maker utility than `MMQ`. Of the policies that have better qualification for both groups and better qualification equality than the decision-maker utility optimal policy, the $\pi_{111}$ parameterization, has the smallest drop in decision-maker utility. Note that $\pi_{111}$ follows the $\pi_{1xx}$ form discussed in Section 4.3.

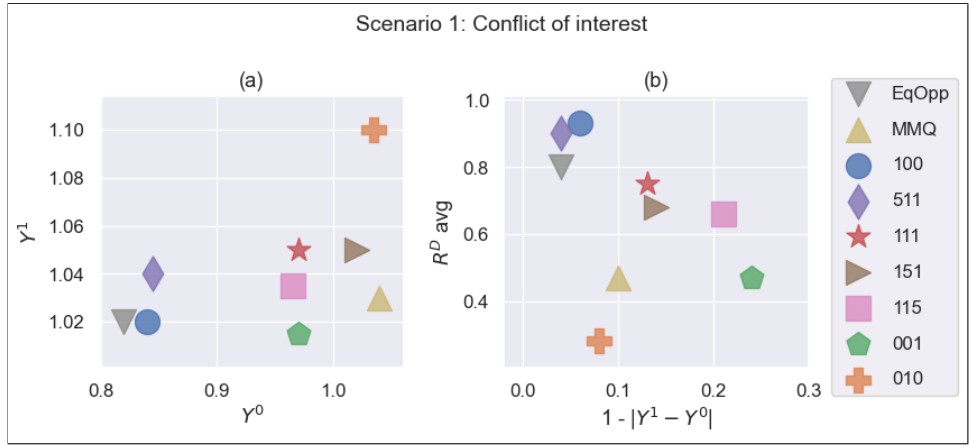

Figure 3: Results of Scenario 1 experiment. Each shape represents the metrics for a particular policy, averaged over 100,000 episodes. Policies labeled with three numbers refer to our approach with the numbers corresponding to $(\lambda^D, \lambda^Q, \lambda^F)$; e.g. `100` corresponds to the decision-maker utility-optimal policy $\pi_{100}$. **(a)**: Average $Y^1$ (advantaged qualification) vs average $Y^0$ (disadvantaged qualification) values. **(b)**: Average $R^D$ (decision-maker utility) vs average $1 - |Y^1 - Y^0|$ (fairness). Axes are aligned so that "up-and-to-the-right" indicates more desirable outcomes.

#### 5.3.2 Scenario 2

Figures 4 show the results of Scenario 2. As shown in Figure 4b, EqOp disrupts the natural fairness that would have been achieved under the decision-maker utility-optimal policy. This is because it requires equality specifically for qualified applicants, which results in inequality when considering all applicants. While a priori this may seem reasonable (adopting EqOp implicitly assumes the effects on the credit scores of unqualified applicants are not necessarily the concern of the bank), as in Experiment 1 we observe that `EqOp` lowers both group qualification averages in order to obtain equality for those considered qualified. Several of the multi-objective approaches ($\pi_{111}, \pi_{115}, \pi_{001}$), on the other hand, improve both the advantaged and disadvantaged

group's qualification. This is a manifestation of Theorem 4.3, which shows that when natural fairness exists, $\lambda^Q \leq \lambda^F$ guarantees qualification improvement to both groups. Interestingly, Figure 4a shows that `MMQ` produces identical qualification values as $\pi_{100}$, [9] but ends up with worse decision-maker utility due to no restriction or decision-maker utility optimization. Again we observe that $\pi_{111}$ provides a strong balance across each objective, and in this scenario we see that it is robust to when the system is naturally fair.

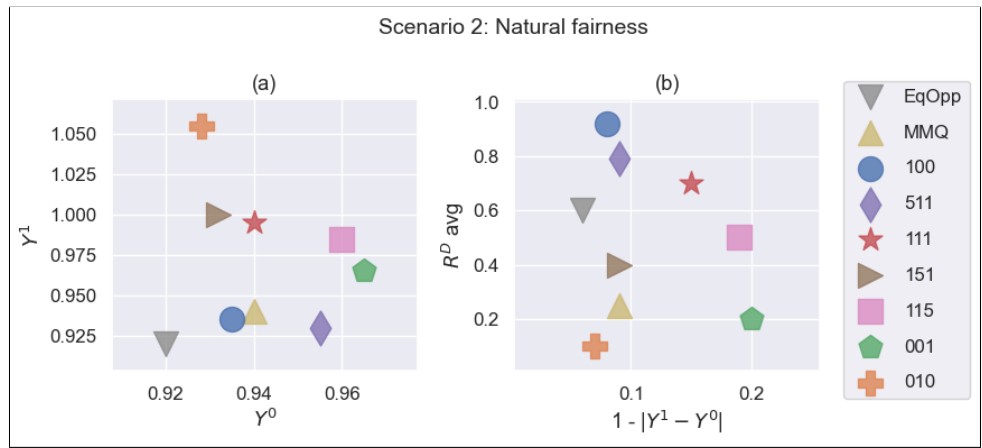

Figure 4: Results of Scenario 2 experiment.

### 5.3.3 Scenario 3

The results of Scenario 3 are shown in Figure 5. Figure 5b shows that `MMQ` has much worse decision-maker utility than the multi-objective approaches with $\lambda^D > 0$. These results support the notion that optimizing for multi-objective rewards produces more balanced policies than maximin approaches which do not make informed trade-offs between the two groups. We also see that `EqOp` violates the no-harm principle by lowering both groups' qualification scores in order to obtain equality.

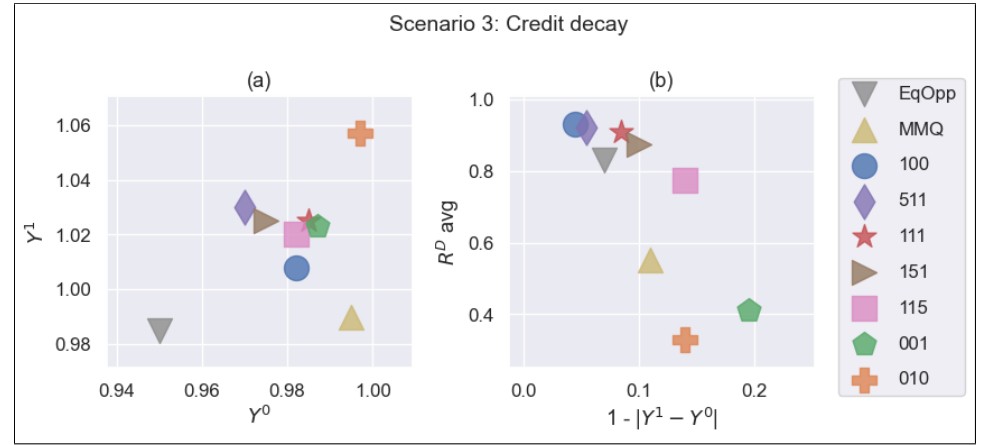

Figure 5: Results of Scenario 3 experiment.

And once again we see that $\pi_{111}$ improves both groups' qualification scores, improves qualification equality, and does so with a minimal drop in decision-maker utility.

---

[9]We added a small amount of noise so that both the `MMQ` and $\pi_{100}$ scatter points were visible on the plot in Figure 4(a).

### 5.4 Default Parameterization

Notice that in Experiments 1-3 that the $\pi_{111}$ policy is not dominated by any other policy on the decision-maker utility vs fairness plots, and is only dominated by the $\pi_{010}$ policy in the qualification plots. This is true only for $\pi_{111}$, which supports our claim that $R^{1xx}$ is a good default parameterization of our approach.

## 6 Conclusion

In this work, we provided a policy-learning solution that achieves group fairness in reinforcement learning. We do so by constructing a multi-objective reward function that balances decision-maker utility, group qualification, and group qualification equality. We demonstrated through mathematical proofs and empirical simulations that our approach is robust to causing harm to one or more groups in order to obtain qualification equality, as well as robust to situations where decision-maker utility and group qualification are distinct entities. Although we restricted our experimental settings to a loan application environment, our model extends to a much broader area of applications where a decision-maker evaluates individuals based on an observed metric, and where the decision itself influences this observed metric in the future. In addition to repeat loan application settings, our model can be extended to criminal recidivism prediction, university admissions decisions, job promotions, predictive policing, and food inspections.

Next we discuss three limitations of our approach. First, our approach assumes that only two groups exist, and that each individual only belongs to a single group. Second, as stated in Section 2, the fairness reward component requires the historical group qualification bias to be present as part of the state, which can cause the MDP to be computationally intractable when learning the optimal policy. Third, our approach requires the existence of a qualification attribute, which may be difficult to observe or define in practice.

Finally, we consider both the positive and negative potential societal impact of our work. On the positive side, our framework gives practitioners confidence that their policies will not harm one or more protected groups. We hope that this confidence will make practitioners more likely to optimize for fairness in practice. As with any fairness framework, however, there is the potential for misuse due to poorly chosen instantiations. A poorly chosen $\lambda$ configuration, for instance, could result in harm to one or more groups.

## Acknowledgments

This material is based upon work supported by the NSF Program on Fairness in AI in Collaboration with Amazon under Award No. 1939743. Any opinion, findings, and conclusions or recommendations expressed in this material are those of the author(s) and do not necessarily reflect the views of the National Science Foundation or Amazon.

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

## A   Full Proofs for Theorems 4.1-4.3

Here we provide full proofs for the theorems in Section 4.1. They show a wider range of conditions suffice than presented in the main text where we focused on the simplest, most natural cases.

*Proof of Theorems 4.1-4.3.* Consider a multi-objective policy $\pi_\lambda$ that is optimal under Equation (1) for some $\lambda$ configuration. In order for $\pi_\lambda$ to do harm, two conditions must hold. First, either $\sum_{\pi_\lambda} Y^0 - \sum_{\pi_{100}} Y^0 < 0$ or $\sum_{\pi_\lambda} Y^1 - \sum_{\pi_{100}} Y^1 < 0$. Second, $\pi_\lambda$ must deviate from the decision-maker utility-optimal policy $\pi_{100}$, which means that

$$\lambda^Q (\sum_{\pi_\lambda} R^Q - \sum_{\pi_{100}} R^Q) + \lambda^F (\sum_{\pi_\lambda} R^F - \sum_{\pi_{100}} R^F) \geq \lambda^D (\sum_{\pi_{100}} R^D - \sum_{\pi_\lambda} R^D). \tag{12}$$

Next we substitute $R^Q$ and $R^F$ values with their $Y^z$ values in Equations (2) and (3):

$$R^Q(\sigma_t) = Y^0_{t+1} + Y^1_{t+1} \quad R^F(\sigma_t) = 1 + \left|Y^\Delta_t\right| - \left|Y^\Delta_{t+1}\right| = 1 + \left|\sum_{i=0}^{t} Y^1_i - \sum_{i=0}^{t} Y^0_i\right| - \left|\sum_{i=0}^{t+1} Y^1_i - \sum_{i=0}^{t+1} Y^0_i\right| \tag{13}$$

to get their summations over an entire episode:

$$\Sigma R^Q(\sigma_t) = \Sigma Y^0_{t+1} + \Sigma Y^1_{t+1}$$

$$\Sigma R^F(\sigma_t) = t + \left|\Sigma Y^1_t - \Sigma Y^0_t\right| - \left|\Sigma Y^1_{t+1} - \Sigma Y^0_{t+1}\right|.$$

When summing over the episode and substituting into Equation 12, the $t$ terms end up canceling since they are present in both the $\sum_{\pi_\lambda} R^F$ and $\sum_{\pi_{100}} R^F$ terms. Also, the $Y^z_t$ terms in Equation (13) telescope when summed over the episode. We are left with:

$$\lambda^Q \left[ (\sum_{\pi_\lambda} Y^0 + \sum_{\pi_\lambda} Y^1) - (\sum_{\pi_{100}} Y^0 + \sum_{\pi_{100}} Y^1) \right] + \lambda^F \left[ (-|\sum_{\pi_\lambda} Y^1 - \sum_{\pi_\lambda} Y^0|) - (-|\sum_{\pi_{100}} Y^1 - \sum_{\pi_{100}} Y^0|) \right] \geq \lambda^D (\sum_{\pi_{100}} R^D - \sum_{\pi_\lambda} R^D). \tag{14}$$

There are two sets of absolute values. By Definition 4.2 and without loss of generality, we know that $\sum\limits_{\pi_{100}} Y^1 - \sum\limits_{\pi_{100}} Y^0 \geq 0$, so we can drop the second set of absolute values. To drop the first set of absolute values, we split the rest of the proofs into two parts, where we prove each of the theorems for the case when the absolute value expression is positive and when it is negative.

## A.1 Part I

Here we prove Theorems 4.1-4.3 under the assumption that the first absolute value expression in Equation (14) is positive. That is, we assume:

$$\sum_{\pi_\lambda} Y^1 \geq \sum_{\pi_\lambda} Y^0 \tag{15}$$

which allows us to simply drop the absolute values:

$$\lambda^Q \left[ \left( \sum_{\pi_\lambda} Y^0 + \sum_{\pi_\lambda} Y^1 \right) - \left( \sum_{\pi_{100}} Y^0 + \sum_{\pi_{100}} Y^1 \right) \right] + \lambda^F \left[ \left( -\left( \sum_{\pi_\lambda} Y^1 - \sum_{\pi_\lambda} Y^0 \right) \right) - \left( -\left( \sum_{\pi_{100}} Y^1 - \sum_{\pi_{100}} Y^0 \right) \right) \right] \geq \lambda^D \left( \sum_{\pi_{100}} R^D - \sum_{\pi_\lambda} R^D \right). \tag{16}$$

After some simple algebra, we get

$$\sum_{\pi_{100}} Y^0 - \sum_{\pi_\lambda} Y^0 \leq \frac{(\lambda^Q - \lambda^F)(\sum_{\pi_\lambda} Y^1 - \sum_{\pi_{100}} Y^1) - \lambda^D (\sum_{\pi_{100}} R^D - \sum_{\pi_\lambda} R^D)}{\lambda^Q + \lambda^F} \tag{17}$$

which serves as an upper bound on the harm to the disadvantaged group. Because no policy can have higher $\Sigma R^D$ than $\pi_{100}$, we know that $\sum\limits_{\pi_{100}} R^D - \sum\limits_{\pi_\lambda} R^D \geq 0$. Therefore, the entire right-hand-side of Equation (17) will be non-positive, which implies no harm to the disadvantaged group, if any of the following conditions are true:

$$\lambda^Q = \lambda^F ; \tag{18a}$$

$$\text{or} \quad \sum_{\pi_\lambda} Y^1 = \sum_{\pi_{100}} Y^1 ; \tag{18b}$$

$$\text{or} \quad (\lambda^Q < \lambda^F) \wedge (\sum_{\pi_\lambda} Y^1 > \sum_{\pi_{100}} Y^1) ; \tag{18c}$$

$$\text{or} \quad (\lambda^Q < \lambda^F) \wedge (\sum_{\pi_\lambda} Y^1 < \sum_{\pi_{100}} Y^1) \tag{18d}$$

$$\wedge \left[ \sum_{\pi_{100}} Y^1 - \sum_{\pi_\lambda} Y^1 \leq \frac{\lambda^D}{\lambda^Q - \lambda^F} \left( \sum_{\pi_{100}} R^D - \sum_{\pi_\lambda} R^D \right) \right] ; \tag{18e}$$

$$\text{or} \quad (\lambda^Q > \lambda^F) \wedge (\sum_{\pi_\lambda} Y^1 \leq \sum_{\pi_{100}} Y^1) ; \tag{18f}$$

$$\text{or} \quad (\lambda^Q > \lambda^F) \wedge (\sum_{\pi_\lambda} Y^1 > \sum_{\pi_{100}} Y^1) \tag{18g}$$

$$\wedge \left[ \sum_{\pi_\lambda} Y^1 - \sum_{\pi_{100}} Y^1 \leq \frac{\lambda^D}{\lambda^Q - \lambda^F} \left( \sum_{\pi_{100}} R^D - \sum_{\pi_\lambda} R^D \right) \right] . \tag{18h}$$

From Equation (18a) we get Theorem 4.1. $\qquad\square$

If we assume that $\lambda^Q > \lambda^F$, we can rearrange the terms in Equation (17) to isolate the harm on the advantaged group to get:

$$\sum_{\pi_{100}} Y^1 - \sum_{\pi_\lambda} Y^1 \leq \frac{(\lambda^Q + \lambda^F)(\sum_{\pi_{100}} Y^0 - \sum_{\pi_\lambda} Y^0) + \lambda^D (\sum_{\pi_{100}} R^D - \sum_{\pi_\lambda} R^D)}{\lambda^F - \lambda^Q}$$

Because no policy can have higher $\Sigma R^D$ than $\pi_{100}$, we know that $\sum\limits_{\pi_{100}} R^D - \sum\limits_{\pi_\lambda} R^D \geq 0$. Therefore, the entire right-hand-side of Equation (A.1) will be non-positive, which implies no harm to the advantaged group, if:

$$(\lambda^Q > \lambda^F) \wedge (\sum_{\pi_{100}} Y^0 \geq \sum_{\pi_\lambda} Y^0) . \tag{19}$$

Combining Equations (19), (18f), and (18a), we see that if $\lambda^Q \geq \lambda^F$ then either the advantaged group or the disadvantaged group is not harmed. This gives us Theorem 4.2. $\qquad\square$

Under natural fairness, $\underset{\pi_{100}}{\Sigma} Y^0 = \underset{\pi_{100}}{\Sigma} Y^1$, so we can replace both values with the same variable $\underset{\pi_{100}}{\Sigma} Y^{0,1} = \underset{\pi_{100}}{\Sigma} Y^0 = \underset{\pi_{100}}{\Sigma} Y^1$. Applying this to Equation (16) yields

$$\lambda^Q \left[ (\underset{\pi_\lambda}{\Sigma} Y^0 + \underset{\pi_\lambda}{\Sigma} Y^1) - 2(\underset{\pi_{100}}{\Sigma} Y^{0,1}) \right] + \lambda^F \left[ (-(\underset{\pi_\lambda}{\Sigma} Y^1 - \underset{\pi_\lambda}{\Sigma} Y^0)) \right] \geq \lambda^D (\underset{\pi_{100}}{\Sigma} R^D - \underset{\pi_\lambda}{\Sigma} R^D) \,,$$

or,

$$\lambda^Q \left[ (\underset{\pi_\lambda}{\Sigma} Y^0 + \underset{\pi_\lambda}{\Sigma} Y^1) - 2 \underset{\pi_{100}}{\Sigma} Y^{0,1} \right] \geq \lambda^D (\underset{\pi_{100}}{\Sigma} R^D - \underset{\pi_\lambda}{\Sigma} R^D) + \lambda^F \left[ \underset{\pi_\lambda}{\Sigma} Y^1 - \underset{\pi_\lambda}{\Sigma} Y^0 \right] \,.$$

By our Equation (15) assumption, the right had side is positive and so $\underset{\pi_\lambda}{\Sigma} Y^1 - \underset{\pi_{100}}{\Sigma} Y^{0,1}$ is positive as well. Similarly, we can simplify Equation (17):

$$\underset{\pi_{100}}{\Sigma} Y^{0,1} - \underset{\pi_\lambda}{\Sigma} Y^0 \leq \frac{(\lambda^Q - \lambda^F)(\underset{\pi_\lambda}{\Sigma} Y^1 - \underset{\pi_{100}}{\Sigma} Y^{0,1}) - \lambda^D (\underset{\pi_{100}}{\Sigma} R^D - \underset{\pi_\lambda}{\Sigma} R^D)}{\lambda^Q + \lambda^F} \,.$$

As before the $R^D$ term is negative so we can drop it and rearrange terms to get:

$$\frac{\underset{\pi_{100}}{\Sigma} Y^{0,1} - \underset{\pi_\lambda}{\Sigma} Y^0}{\underset{\pi_\lambda}{\Sigma} Y^1 - \underset{\pi_{100}}{\Sigma} Y^{0,1}} \leq \frac{\lambda^Q - \lambda^F}{\lambda^Q + \lambda^F} \,,$$

or:

$$\frac{\underset{\pi_\lambda}{\Sigma} Y^0 - \underset{\pi_{100}}{\Sigma} Y^{0,1}}{\underset{\pi_\lambda}{\Sigma} Y^1 - \underset{\pi_{100}}{\Sigma} Y^{0,1}} \geq \frac{\lambda^F - \lambda^Q}{\lambda^Q + \lambda^F} \,.$$

If $\lambda^F \geq \lambda^Q$, then the left-hand side must be non-negative. This implies that $\underset{\pi_\lambda}{\Sigma} Y^0 - \underset{\pi_{100}}{\Sigma} Y^{0,1}$ is also non-negative, which gives us Theorem 4.3. □

## A.2 Part II

Here we prove Theorems 4.1-4.3 under the assumption that the first absolute value expression in Equation (14) is negative. That is, we assume:

$$\underset{\pi_\lambda}{\Sigma} Y^0 \geq \underset{\pi_\lambda}{\Sigma} Y^1 \tag{20}$$

which gives us:

$$\lambda^Q \left[ (\underset{\pi_\lambda}{\Sigma} Y^0 + \underset{\pi_\lambda}{\Sigma} Y^1) - (\underset{\pi_{100}}{\Sigma} Y^0 + \underset{\pi_{100}}{\Sigma} Y^1) \right] + \lambda^F \left[ (\underset{\pi_\lambda}{\Sigma} Y^1 - \underset{\pi_\lambda}{\Sigma} Y^0) - (-(\underset{\pi_{100}}{\Sigma} Y^1 - \underset{\pi_{100}}{\Sigma} Y^0)) \right] \geq \lambda^D (\underset{\pi_{100}}{\Sigma} R^D - \underset{\pi_\lambda}{\Sigma} R^D) \,. \tag{21}$$

After some simple algebra we get:

$$(\lambda^Q + \lambda^F)(\underset{\pi_{100}}{\Sigma} Y^0 - \underset{\pi_\lambda}{\Sigma} Y^1) \leq (\lambda^F - \lambda^Q)(\underset{\pi_{100}}{\Sigma} Y^1 - \underset{\pi_\lambda}{\Sigma} Y^0) - \lambda^D (\underset{\pi_{100}}{\Sigma} R^D - \underset{\pi_\lambda}{\Sigma} R^D) \,. \tag{22}$$

As before, we know that $\underset{\pi_{100}}{\Sigma} R^D - \underset{\pi_\lambda}{\Sigma} R^D \geq 0$, so

$$(\lambda^Q + \lambda^F)(\underset{\pi_{100}}{\Sigma} Y^0 - \underset{\pi_\lambda}{\Sigma} Y^1) \leq (\lambda^F - \lambda^Q)(\underset{\pi_{100}}{\Sigma} Y^1 - \underset{\pi_\lambda}{\Sigma} Y^0) \,. \tag{23}$$

If $\lambda^Q = \lambda^F$, then

$$(\lambda^Q + \lambda^F)(\underset{\pi_{100}}{\Sigma} Y^0 - \underset{\pi_\lambda}{\Sigma} Y^1) \leq 0 \tag{24}$$

so $\underset{\pi_\lambda}{\Sigma} Y^1 \geq \underset{\pi_{100}}{\Sigma} Y^0$. Combining this with the Equation (20) gives us

$$\underset{\pi_\lambda}{\Sigma} Y^0 \geq \underset{\pi_\lambda}{\Sigma} Y^1 \geq \underset{\pi_{100}}{\Sigma} Y^0 \,.$$

This is Theorem 4.1. □

If $\lambda^Q > \lambda^F$ then the right-hand-side of Equation (23) is negative if $\underset{\pi_{100}}{\Sigma} Y^1 > \underset{\pi_\lambda}{\Sigma} Y^0$. When the right-hand-side is negative, then the left-hand-side must also be negative, which implies that $\underset{\pi_\lambda}{\Sigma} Y^1 > \underset{\pi_{100}}{\Sigma} Y^0$. From this and the Equation 20 assumption, we get

$$\underset{\pi_\lambda}{\Sigma} Y^0 \geq \underset{\pi_\lambda}{\Sigma} Y^1 > \underset{\pi_{100}}{\Sigma} Y^0 \,. \tag{25}$$

Alternatively, if the right-hand-side of Equation (23) is positive, then $\underset{\pi_\lambda}{\Sigma} Y^0 > \underset{\pi_{100}}{\Sigma} Y^1$. Combining this with Definition 4.2, we have

$$\begin{aligned} \underset{\pi_\lambda}{\Sigma} Y^0 &> \underset{\pi_{100}}{\Sigma} Y^1 \,, \\ \underset{\pi_{100}}{\Sigma} Y^1 &> \underset{\pi_{100}}{\Sigma} Y^0 \end{aligned} \tag{26}$$

which gives us $\underset{\pi_\lambda}{\Sigma} Y^0 > \underset{\pi_{100}}{\Sigma} Y^0$. Combining this with Equation (25) gives us Theorem 4.2. $\qquad\square$

Under natural fairness, $\underset{\pi_{100}}{\Sigma} Y^0 = \underset{\pi_{100}}{\Sigma} Y^1 \equiv \underset{\pi_{100}}{\Sigma} Y^{0,1}$. Applying this to Equation (21) yields

$$\lambda^Q \big[ (\underset{\pi_\lambda}{\Sigma} Y^0 + \underset{\pi_\lambda}{\Sigma} Y^1) - (\underset{\pi_{100}}{\Sigma} Y^{0,1} + \underset{\pi_{100}}{\Sigma} Y^{0,1}) \big] + \lambda^F \big[ (\underset{\pi_\lambda}{\Sigma} Y^1 - \underset{\pi_\lambda}{\Sigma} Y^0) - (-(\underset{\pi_{100}}{\Sigma} Y^{0,1} - \underset{\pi_{100}}{\Sigma} Y^{0,1})) \big] \geq \lambda^D (\underset{\pi_{100}}{\Sigma} R^D - \underset{\pi_\lambda}{\Sigma} R^D) \,.$$

Simplifying yields

$$\lambda^Q \big[ (\underset{\pi_\lambda}{\Sigma} Y^0 + \underset{\pi_\lambda}{\Sigma} Y^1) - (\underset{\pi_{100}}{\Sigma} Y^{0,1} + \underset{\pi_{100}}{\Sigma} Y^{0,1}) \big] \geq \lambda^F \big[ \underset{\pi_\lambda}{\Sigma} Y^0 - \underset{\pi_\lambda}{\Sigma} Y^1 \big] \,.$$

Suppose for contradiction group 1 is harmed. Then

$$\lambda^Q \big[ \underset{\pi_\lambda}{\Sigma} Y^0 - \underset{\pi_{100}}{\Sigma} Y^{0,1} \big] > \lambda^F \big[ \underset{\pi_\lambda}{\Sigma} Y^0 - \underset{\pi_\lambda}{\Sigma} Y^1 \big] \geq \lambda^F \big[ \underset{\pi_\lambda}{\Sigma} Y^0 - \underset{\pi_{100}}{\Sigma} Y^{0,1} \big] \,.$$

If $\lambda^F \geq \lambda^Q$ this is a contradiction, which gives us Theorem 4.3. $\qquad\square$

# B Relative Harm Properties

In Section 4 we provided definitions and theorems based on *harm*, where harm is defined with the decision-maker utility-optimal policy $\pi_{100}$ as the baseline policy. We can, however, generalize our definition of harm as well as our theorems by defining *relative* harm. Instead of using $\pi_{100}$ as the baseline, *relative harm* uses some multi-objective policy $\pi_{\lambda'}$ as the baseline policy.

**Definition B.1** (Relative Harm). *With respect to some baseline policy $\pi'$, a policy $\pi$ does **relative harm** to group $Z = z$ if the group's expected cumulative qualification value is lower under $\pi$ than under $\pi'$:*

$$\underset{\pi'}{\Sigma} Y^z > \underset{\pi}{\Sigma} Y^z \,. \tag{27}$$

*Similarly, we say that a policy $\pi$ does **no relative harm** to group $Z = z$ if $\underset{\pi'}{\Sigma} Y^z \leq \underset{\pi}{\Sigma} Y^z$ and **relatively benefits** group $Z = z$ if $\underset{\pi'}{\Sigma} Y^z < \underset{\pi}{\Sigma} Y^z$.*

We can reconstruct our Section 4.1 properties with respect to *relative harm* by adding the condition that there is no improvement to decision-maker utility and by defining the advantaged/disadvantaged groups with respect to the baseline multi-objective policy $\pi_{\lambda'}$ Without loss of generality, we set the group that is disadvantaged under the baseline policy $\pi_{\lambda'}$ as $Z = 0$. We can guarantee the following properties for multi-objective policies $\pi_\lambda$ that have the specified changes in $\lambda$ relative to their $\lambda'$ values. The proofs are identical to their Section 4.1 counterparts after replacing $\pi_{100}$ with $\pi_{\lambda'}$ and adding the constraint that there is no improvement to decision-maker utility.

**Theorem B.1.** *Relative to some policy $\pi_{\lambda'}$ where $\lambda' = (\lambda^{Q'}, \lambda^{F'}, \lambda^{D'})$, and when natural fairness does not exist, no relative harm is done to the disadvantaged group if $\lambda^Q - \lambda^{Q'} = \lambda^F - \lambda^{F'}$ and $\underset{\pi_{\lambda'}}{\Sigma} R^D \geq \underset{\pi_\lambda}{\Sigma} R^D$.*

This is the relative form of Theorem 4.1. Intuitively, it says that no relative harm is done to the disadvantaged group if the change in $\lambda^Q$ is equal to the change in $\lambda^F$ and there is no improvement to decision-maker utility.

**Theorem B.2.** *Relative to some policy $\pi_{\lambda'}$, at least one group is not relatively harmed if $\lambda^Q - \lambda^{Q'} \geq \lambda^F - \lambda^{F'}$ and $\sum_{\pi_{\lambda'}} R^D \geq \sum_{\pi_\lambda} R^D$.*

This is the relative form of Theorem 4.2.

**Theorem B.3.** *Relative to some policy $\pi_{\lambda'}$, when $\sum_{\pi_{\lambda'}} Y^z = \sum_{\pi_{\lambda'}} Y^{1-z}$, neither group is relatively harmed if $\lambda^Q - \lambda^{Q'} \leq \lambda^F - \lambda^{F'}$ and $\sum_{\pi_{\lambda'}} R^D \geq \sum_{\pi_\lambda} R^D$.*

This is the relative form of Theorem 4.3. The $\sum_{\pi_{\lambda'}} Y^z = \sum_{\pi_{\lambda'}} Y^{1-z}$ constraint is similar to natural fairness except that it is for some baseline $\pi_{\lambda'}$ rather than $\pi_{100}$.

## C Harm Properties with Approximate Solutions

In this appendix we explain how our results guaranteeing no harm apply in settings where we find only approximately optimal policies. In particular, we show how an additive approximation error can be carried through our analysis, providing a bound on harm that is linear in the approximation error.

Our analysis for the case of exact optimization came from the observation that

$$\lambda^Q\left(\sum_{\pi_\lambda} R^Q - \sum_{\pi_{100}} R^Q\right) + \lambda^F\left(\sum_{\pi_\lambda} R^F - \sum_{\pi_{100}} R^F\right) \geq \lambda^D\left(\sum_{\pi_{100}} R^D - \sum_{\pi_\lambda} R^D\right). \tag{28}$$

This came from $\pi_\lambda$ being optimal, meaning in particular it must (weakly) improve on $\pi_{100}$. If instead it is optimized to within an additive error of $\epsilon$ we would have

$$\lambda^Q\left(\sum_{\pi_\lambda} R^Q - \sum_{\pi_{100}} R^Q\right) + \lambda^F\left(\sum_{\pi_\lambda} R^F - \sum_{\pi_{100}} R^F\right) + \epsilon \geq \lambda^D\left(\sum_{\pi_{100}} R^D - \sum_{\pi_\lambda} R^D\right). \tag{29}$$

After some simple algebra, instead of (17) we get

$$\sum_{\pi_{100}} Y^0 - \sum_{\pi_\lambda} Y^0 \leq \frac{(\lambda^Q - \lambda^F)\left(\sum_{\pi_\lambda} Y^1 - \sum_{\pi_{100}} Y^1\right) - \lambda^D\left(\sum_{\pi_{100}} R^D - \sum_{\pi_\lambda} R^D\right) + \epsilon}{\lambda^Q + \lambda^F} \tag{30}$$

Furthermore, since $\pi_{100}$ may also be optimal only up to an additive factor of $\epsilon$, the term $\left(\sum_{\pi_{100}} R^D - \sum_{\pi_\lambda} R^D\right)$ may be negative by up to $\epsilon$. Thus, rather than concluding that the harm is non-positive we can only conclude it is at most $\frac{(1+\lambda^D)\epsilon}{\lambda^Q + \lambda^F}$. Thus, the possible harm grows linearly in the approximation error. Furthermore, we can see that increasing the weight on fairness (i.e., taking $\lambda^Q = \lambda^F$ as discussed in Section 4.3 and increasing this hyperparameter) decreases the possible harm.

Note that this bound goes to infinity as we take the weight on fairness to 0. This is intuitive. If we were not interested in fairness and simply optimized $\pi_{100}$ twice, while the objective (decision-maker utility) can only be harmed by $\epsilon$ due to approximation error, this could cause arbitrary changes to the qualification of different groups. Putting weight on fairness is what allows us to translate bounds on approximation error into bounds on harm.

## D Extension to More than Two Groups

Our main focus is on the two-group setting, $|\mathcal{Z}| = 2$, which reflects the most common scenario in group fairness. We can, however, generalize our approach to more than two groups as follows.

We keep the overall reward structure consistent with Equation (1), which we repeat here for convenience:

$$R(\lambda, \sigma_t) = \lambda^D R^D(\sigma_t) + \lambda^Q R^Q(\sigma_t) + \lambda^F R^F(\sigma_t),$$

but we define $R^Q$ and $R^F$ as functions of $|\mathcal{Z}|$:

$$R^Q(\sigma_t) = \frac{1}{|\mathcal{Z}|} \sum_{i=0}^{|\mathcal{Z}|} Y_{t+1}^i, \tag{31}$$

$$R^F(\sigma_t) = \frac{1}{|\mathcal{Z}|}\left(1 + \sum_{i=0}^{|\mathcal{Z}|}\left|Y_t^i - Y_t^{\text{avg}}\right| - \sum_{i=0}^{|\mathcal{Z}|}\left|Y_{t+1}^i - Y_{t+1}^{\text{avg}}\right|\right) \tag{32}$$

where

$$Y_t^{\text{avg}} = \frac{1}{|\mathcal{Z}|}\sum_{i=0}^{|\mathcal{Z}|}Y_t^i. \tag{33}$$

Under this scenario, our theoretical properties need to be adjusted in order to consider that there is no longer a clear "disadvantaged" and "advantaged" group. A full theoretical analysis of this new case likely needs new ideas, as the two cases we consider in our proof become twelve cases, even with only three protected groups. However, we provide numerical illustrations in Figure 6 to show how our approach is still relevant in this more general setting. As we can see, similar to the two-group setting, $\lambda^Q < \lambda^F$ narrows the band of qualification values to only consider qualification combinations more optimal if they are all close together. Similarly, $\lambda^Q = \lambda^F$ creates a reward where higher $Y$ values are considered better, but only if it is not at the expense of lowering another group's qualification, as characterized by the box-like contours upward and rightward from the baseline. Finally, $\lambda^Q > \lambda^F$ still encourages higher qualification values, even if it is at the expense of lowering another group's qualification.

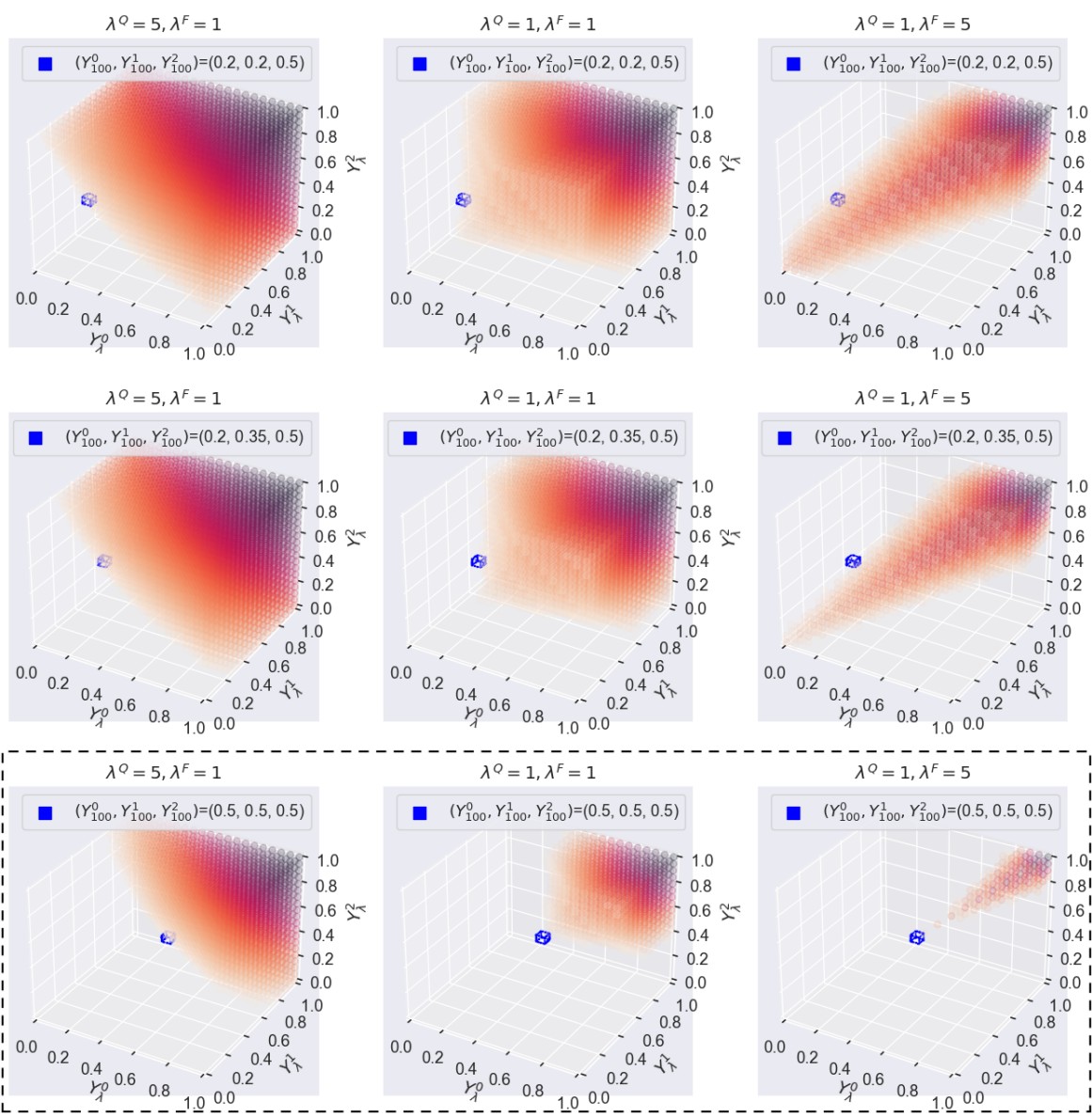

Figure 6: Plots showing the "desirability" (i.e. $\Delta R^{QF}$) of each possible $(Y_\lambda^0, Y_\lambda^1)$ outcome for three different decision-maker utility-optimal $Y$ values and three different $\lambda^Q/\lambda^F$ ratios. The blue cubes indicate the $Y$ values for the decision-maker utility-optimal policy, which will by definition have $\Delta R^{QF} = 0$. Darker contours indicate more attractive (higher $\Delta R^{QF}$) values. The colorless areas thus indicate where $\Delta R^{QF} \leq 0$, which means they are not possible solutions under $\pi_\lambda$ since the decision-maker utility-optimal policy will be preferred over them. The dotted area indicates the scenarios with natural fairness. The plots assume $\Delta R^D = 0$ so that decision-maker utility is still optimal and any $\Delta R^\lambda$ originates exclusively from $\Delta R^Q$ and $\Delta R^F$.

