# OpenReview forum: "Group Fairness in Reinforcement Learning via Multi-Objective Rewards"
_TMLR — Accepted by TMLR_

### Review · Reviewer_hMiN · 2024-01-18

**Summary Of Contributions:**

The paper addresses the issue of ensuring that the sequential decision-making policy learned by a reinforcement learning agent satisfies fairness requirements, specifically in terms of fairness between groups. It does this via a multi-objective formulation of the reward, with separate terms accounting corresponding to decision maker-utility, group qualification, and group qualification equality. These are combined via a weighted sum process to form a scalar reward (which can therefore be optimised for using standard RL methods). The authors show both theoretically and via a small empirical example that appropriately chosen weights can achieve suitable trade-offs between the different objectives. In particular this approach avoids violations of the “no harm” principle (ie it does not allow one group’s qualification to be lowered in order to improve group qualification equality), and allows for a suitable balancing of decision-maker utility and group qualification – previous approaches to fair RL have not provided this functionality.

**Audience:**

Yes

**Broader Impact Concerns:**

I don’t have any further concerns beyond those which the authors have already addressed in their Conclusion. Including fairness objectives in addition to decision-maker utility clearly offers the potential for more ethical development and deployment or RL systems than is the case for standard RL methods that focus exclusively on decision-maker utility.

**Claims And Evidence:**

Yes

**Requested Changes:**

A clearer explanation of the MDP’s ‘no cash’ situation and of the process used to produce the results in Fig 3 is required, and would be important for me in recommending acceptance as the results are possibly open to misinterpretation without that.

I would also like to see the authors address the other issues I have identified above (briefly summarised below), but these re less critical – more a matter of making the paper easier to read and more impactful rather than of meeting the criteria for acceptance
-	Add worked examples in section 4.1
-	Discussion of the results of policies 511 vs 100
-	Extended discussion of limitations and possible extensions in the Conclusion

There are also some smaller presentation issues which should be addressed in the final copy:
-	On p5 the statement is made that “R^D is the decision-maker utility function defined in Section 2”. But Section 2 doesn’t actually define R^D – it just says the lender wants to optimize profit).
-	Also on p5 R^Q is described as the *average* of group qualifications, whereas it is actually the *sum* of group qualifications.
-	P6 starts with a duplicated a: “a a single objective”
-	Fig 2: The labels for the Y100 values superimposed on each graph are very hard to read – small font on a grey background. I had to zoom in a long way on the pdf before these were legible, and this contributed to the time it took me to understand this figure.
-	P8: “thee” should be “the”

**Strengths And Weaknesses:**

The paper addresses an important problem of ensuring that the policies followed by reinforcement learning systems account for fairness requirements rather than simply maximising decision-maker utility. The authors have identified significant limitations in prior approaches to fair RL, and have developed an approach which addresses those limitations.

The approach itself is elegantly simple – three rewards addressing different objectives which we wish to optimise, which are combined together via a weighted scalarisation to produce a scalar reward. As a multiobjective RL researcher, I was initially concerned by the use of a linear weighted sum of objectives, as it’s well known in MORL that this may be incapable of finding some Pareto-optimal solutions. But the authors do a good job of demonstrating that given the interconnected nature of the second and third objectives, a linear weighting makes sense. It simplifies analysis of the effect of changing the weighting parameters, and also means that this technique can be implemented using any existing single-objective RL method, which is practically important given that these algorithms are generally more advanced and better understood than their MORL counterparts.

The formal analysis of the effect of the weights and the empirical demonstration in Section 5 do a good job of highlighting the benefits of this new approach compared to the previous fair RL methods which are used as baselines. (I should note that my mathematical expertise is not high, so I can’t vouch for the correctness of the proofs, but I studied them extensively and did not see any flaws).
Overall I thought the paper was well-written and clearly presented. In particular the first 3 sections really clearly defined the problem area and the proposed approach. I did find the Analysis section much harder to follow – I had to re-read this multiple times. It took me a while to comprehend Fig 2, but once I did I found this assisted my understanding of the earlier material, so I think it might be beneficial to provide some worked numeric examples in Section 4.1 as well.

The main weakness of the proposed approach is that it is restricted to problems involving just two groups, whereas in practice we would usually require policies to be fair across multiple protected attributes. The authors do acknowledge this limitation in the conclusion, and an approach which works for the special case of two groups is still of value, but I would have expected to see some discussion about whether this approach might be extended to the case of >2 groups, or whether an entirely different method will be required.

In the conclusion the authors also note a limitation in that the inclusion of the historical group qualification bias as part of the state may make learning the optimal policy from the MDP computationally intractable. In the empirical study, the authors have used linear programming to find the optimal policy (which to my understanding requires a complete model of the MDP, rather than learning from experience like most RL methods – please correct me if my assumption here is incorrect). Given that the paper is framed in terms of RL, I would have preferred to see results based on a model-free method, or at least some discussion of whether standard model-free RL could be applied.

I also had some difficulty in understanding the nature of the MDP used in Section 5, and how it was executed in the context of the empirical evaluation. The cash attribute of the lender is in the range (0,1,2) starting with a value of 1, and increases or decreases by 1 (capped to that range) whenever a loan is paid back or defaulted. For even the most qualified lendees, the default rate is 25%. So it seems to me that even following an optimal policy, executing this MDP will almost inevitably result in entering the ‘no cash’ state (only 2 defaults in a row are required to go from the maximum cash value of 2 to the minimum of 0), and once this state is entered it can not be exited as the lender can not make any further loans. The results in Fig 3 are reported as being over 10,000 episodes – it seems inevitable that the ‘no cash’ situation arises during that time-period, so how is this handled?

Looking more closely at the results in Fig 3, it is surprising that policy 511 outperforms 100 on the decision-maker utility in Scenario 1, and dominates it in Scenario 3. 100 focuses exclusively on the decision-maker utility, whereas 511 applies at least some weighting to the fairness objectives, so it seems unusual that it does better with regards to the decision-maker utility? Is this just due to the stochasticity of the MDP (in which case perhaps more than 10K episodes are required), or is there some other algorithmic factor at play here?

Finally, while I earlier identified the choice to scalarise the objectives into a single reward as a strength (as it allows standard single-objective RL methods to be used), this is also a potential limitation of the approach, as it eliminates the possibility of carrying out multi-policy learning. Even if we limit the weight-space to the 1xx option recommended by the authors, we may need to execute the learning process multiple times with different values of x in order to find a policy which provides a suitable trade-off between decision-maker utility and the fairness objectives (as is discussed in Section 4.3). For a simple MDP and the linear programming method used here, the cost of finding several policies is not excessive, but for more complex MDPs and learning methods which actually require sampled experience from the environment this could become costly. If the problem was treated as a MOMDP, then multi-policy MORL methods could be used to learn multiple policies in parallel (for example, the Conditioned Network algorithm of Abels et al, 2019 can learn multiple linear-weighted policies, so could be a good fit for this task).

Reference
Abels, A., Roijers, D., Lenaerts, T., Nowé, A., & Steckelmacher, D. (2019, May). Dynamic weights in multi-objective deep reinforcement learning. In International conference on machine learning (pp. 11-20). PMLR.

---

> ### Author Response · Authors · 2024-03-06
> **Response to feedback and corresponding changes to our revision**
>
> Thank you for your insightful and constructive feedback. We appreciate the opportunity to address the concerns raised and outline the improvements we've made based on your comments.
>
> **Clarifications on MDP and Experiment Setup**
>
> > "A clearer explanation of the MDP’s ‘no cash’ situation and of the process used to produce the results in Fig 3 is required, and would be important for me in recommending acceptance as the results are possibly open to misinterpretation without that."
>
> We acknowledge the need for a clearer explanation of the MDP, including a more detailed explanation of how the results were generated in Figure 3.   We have revised the presentation in Section 5, including a new subsection 5.1.1.  explaining and motivating the scenarios in a single location.
>
> Referring to your comment,
> > "The results are in Fig 3 are reported as being over 10,000 episodes - it seems inevitable that the 'no cash' situation arises during that time period."
>
> This is correct. The no-cash situation certainly does arise, as intended. In this situation, the lender has no cash to give out loans, and they have no way of acquiring more cash for future loans. However,  note that while we ran 10,000 episodes (now 100,000; see below), in each episode of our two-stage loan application model, there are only two timesteps, which we have made explicit in our revision.
>
> **Extended Discussion of Limitations and Possible Extensions**
>
> > "Extended discussion of limitations and possible extensions"
>
> Your point about the limitation of our approach to problems involving just two groups is well-taken, and we've provided a new Appendix section  (Appendix D) that discusses the extension to more than two groups.
>
> **Improvements to understanding Figure 2**:
>
> >  "Add worked examples in section 4.1"
>
> To address the difficulty in understanding Figure 2, we've added a worked example to Section 4.2 and included helpful tags to the plot (e.g., "natural fairness" in the bottom row, and which plots illustrate which theorems).
>
> **Strengthening Empirical Evidence**
>
> >  "Discussion of the results of policies 511 vs 100"
>
> In response to your concern about stochasticity and sampling error, particularly regarding the performance of the 511 vs. 100 configurations,  we increased the episode count in our experiments from 10,000 to 100,000.  Now the sampling error is sufficiently small that we see 100 having optimal R^D in all 3 scenarios.
>
> **Scalar Reward and Multi-Policy Learning**
>
> > "Finally, while I earlier identified the choice to scalarise the objectives into a single reward as a strength (as it allows standard single-objective RL methods to be used), this is also a potential limitation of the approach, as it eliminates the possibility of carrying out multi-policy learning. Even if we limit the weight-space to the 1xx option recommended by the authors, we may need to execute the learning process multiple times with different values of x in order to find a policy which provides a suitable trade-off between decision-maker utility and the fairness objectives (as is discussed in Section 4.3). For a simple MDP and the linear programming method used here, the cost of finding several policies is not excessive, but for more complex MDPs and learning methods which actually require sampled experience from the environment this could become costly. If the problem was treated as a MOMDP, then multi-policy MORL methods could be used to learn multiple policies in parallel (for example, the Conditioned Network algorithm of Abels et al, 2019 can learn multiple linear-weighted policies, so could be a good fit for this task)."
>
> We acknowledge and thank the reviewer for their critique regarding the scalar reward framework potentially limiting the exploration of multi-policy learning. In response, we've included a paragraph at the end of Section 4.3 that discusses the MOMDP vs scalar reward tradeoff.
>
> We have also addressed the minor presentation issues pointed out, e.g. making clear that the notation R^D rather than the function itself was defined in section 3.
>
>
> Thank you for your thoughtful review and suggestions. We look forward to your further feedback.

---

> > ### Comment · Reviewer_hMiN · 2024-03-08
> > **Response to revision**
> >
> > I am satisfied by the revisions made to address my original comments. In particular I have a much clearer understanding of the structure of the experiment (100,000 episodes of 2-steps each, where cash is set to 1 at the start of each episode) than I did after reading the original submission.

---

### Review · Reviewer_fmFj · 2024-02-05

**Summary Of Contributions:**

The paper looks at the problem of group fairness in the RL setting. They approach this problem via proposing three implicit objectives which are then converted to a single scalar objective that can be solved with any traditional scalar RL methods. The three objectives are based on: maximizing the system-designer/decision-maker's utility, maximizing the group's utilities (qualification) and minimizing the difference between groups. These objectives are then scaled via some hyper-parameters and then combined together. The authors claim that the proposed method addresses the drawbacks of other approaches, i.e., they can explicitly model the decision-maker's utility and ensure the "no-harm" property. Here no-harm property refers that the method prevents drop in utility of a group on introducing fairness. They provide empirical evidence of their method on a small synthetic MDP lending task.

**Audience:**

Yes

**Broader Impact Concerns:**

Please see my comment about limitations in the above section.

**Claims And Evidence:**

No

**Requested Changes:**

- Notation: Is the paper in the context of infinite-horizon discounted MDPs or finite horizon. Both the notations are used throughout the paper which can be confusing at times.
- Clarity: Both Fig 2 and 3 pack too much information and are tough to read. I would encourage the authors to only keep the most representative figures for readability. For Fig 3, the authors can also use different markers along with different colors.
- Empirical studies: The paper needs more evidence to support their claims, particularly of those regarding the drawbacks. The authors can  use other studies like [3] or some DeepRL experiments.
- Analysis: There should be some discussion of the trade-offs with $R^D$ (w.r.t. $\pi_{100}$?). Additionally, more discussion should be on how to find the weights so that there is "no-harm" to both the groups. If this can not be achieved, then the authors should mention that too.
- Limitations: More emphasis should be placed the the method is limited to only two groups. Additionally, when the MDP is not known (RL setting), even setting the correct weights might lead to harm when the learning algorithm is still getting information about the environment [1,2].


### References:
- [3] Dynamic Positive Reinforcement For Long-Term Fairness, Puranik et al, ICLR 2022 Workshop on Socially Responsible Machine Learning.

**Strengths And Weaknesses:**

## Strengths

- The problem of fairness in RL is an important one. The authors provide an alternate formulation of this problem that includes the utilities of the groups and differences between them in the modelling itself (graphical model in Fig 1).
- The paper is written clearly and easy to read.

## Weaknesses
- The methodology is limited to only two groups. It is not clear how the approach would scale with number of groups. For instance, I believe there will be combinatorial hyperparameters or weights for including each pairwise term.
- The core contribution of the paper is the weighted-sum objective (Eq 1). Similar objectives have considered in the literature [1, 2] where the weighted-sum objective can be considered to the lagrangian versions of the objectives studied in the related papers. The main claims of the work is how the authors claim their proposed objective can address the "no-harm" drawback. However, this claim is not supported in the analysis or the results. For instance, when no natural fairness exists, i.e., there is discrepancy between groups, Theorem 4.1 is only provide guarantees for the disadvantaged group (but not for advantageous group). Similarly, Thm 4.2 just provides results that at least one group (out of 2) is not harmed. In both the cases, there is case for harm for one out of the two groups, so it is not clear how the proposed approach is better than the existing work in terms of drawbacks.
- There is little discussion on how the weights should be set, especially for the non-natural fairness case. If the author's are proposing to always increase while satisfying $\lambda^Q = \lambda^F$, then shouldn't the analysis be done w.r.t. this objective.
- There is limited empirical evidence of the claims. All the experiments are done a small synthetic lending environment where the behaviour seems contrived. For instance, in Sec 5.3.1, when making comparison to EqOp baseline, all the proposed weights/approaches that do not violate the no-harm principle achieve a lower $R^D$ than the EqOp baseline. There is clear trade-off here which is not discussed in the paper.



### References:
- [1] Algorithms for fairness in sequential decision making, Wen et al, AISTATS 2021.
- [2] Group fairness in RL, Satija et al, TMLR 2023.

---

> ### Author Response · Authors · 2024-03-06
> **Response to feedback and corresponding changes to our revision**
>
> Thank you for your insightful and constructive feedback. Below we address the concerns raised and outline the improvements we've made based on your comments.
>
> ## Finite vs infinite-horizon MDPs
>
> > "Is the paper in the context of infinite-horizon discounted MDPs or finite horizon."
>
> Thank you for pointing out this discrepancy. We've updated our notation to keep it consistent with the infinite horizon setting––Unnumbered equation at the end of Section 2, Equation (4), and Equation (10).
>
> ## Notation and Figure Clarity
>
> > "Both Fig 2 and 3 ... tough to read... For Fig 3, the authors can also use different markers..."
>
> We agree that Figures 2 and 3 could benefit from enhanced readability. We have revised these figures to help with clarity.
>
> ## Discussion on Trade-Offs and Finding Weights
>
> > " Additionally, more discussion should be on how to find the weights so that there is "no-harm" to both the groups."
>
> We discuss this issue some in 4.3, the space of parameters we offer is not so large that a practitioner cannot use their own strategies for selecting the appropriate configurations. As discussed in below, the goal is not always to achieve no-harm to both groups, although when possible that is of course desirable.
>
> ## Clarification on Weighted-Sum Objective and "No-Harm" Property:
>
> > “Similar objectives have considered in the literature [1, 2] where the weighted-sum objective can be considered to the lagrangian versions ...main claims ...no-harm" drawback. However, this claim is not supported....when no natural fairness exists, i.e., there is discrepancy between groups, Theorem 4.1 is only provide guarantees for the disadvantaged group... Thm 4.2 just provides results that at least one group..."
>
> Thank you for your feedback on the lack of clarity regarding our weighted-sum objective and its connection to the "no-harm" property. However, there seems to be a misunderstanding of our objective, for which we must take responsibility due to our use of unclear terminology. To clarify, the problem we are trying to solve with respect to 'harm' is that both groups are not lowered as a result of trying to achieve fairness, which comes from the use of constraints. Your interpretation, suggesting no harm occurs under either group, can be overly restrictive in that it is only applicable under very specific circumstances. Instead, our focus is on assessing the trade-offs involved, specifically how the qualification standards for one group may be adjusted in favor of another. We have made substantial revisions to 1.1 and 1.2 that better define our objective, including with respect to harm.
>
> ## Empirical Evidence and Methodology Clarity
>
> > "Empirical studies: The paper needs more evidence to support their claims, particularly of those regarding the drawbacks. The authors can use other studies like [3] or some DeepRL experiments."
>
> In our revision, we have clarified the goals of our experiments at the start of Section 5, and don’t believe additional experiments are needed to achieve them.  Our choice of setting to study is quite consistent with prior work in this space that also typically does not do the sort of large-scale deep RL experiments suggested.
>
> > “For instance, in Sec 5.3.1, when making comparison to EqOp baseline, all the proposed weights/approaches that do not violate the no-harm principle achieve a lower R^D than the EqOp baseline. There is clear trade-off here which is not discussed in the paper.”
>
> As the revised version of the figures hopefully makes clearer, this is not an interesting tradeoff because, as discussed at the start of the subsection and shown in the left subfigure EqOp is simply harming both groups in the name of “fairness”. Since we don’t believe there is a meaningful fairness gain here, the decision maker would be better off with the 100 policy which does not consider fairness at all. There is indeed a tradeoff between that and the fair policies, which is explicitly discussed.
>
> ## Addressing the Limitation to Two Groups:
>
> > "Limitations: More emphasis should be placed the the method is limited to only two groups."
>
> We acknowledge the limitation and the potential combinatorial complexity in scaling to multiple groups. In response, we added discussion on this aspect, exploring theoretical extensions and potential methodologies for scaling to more groups without exponentially increasing complexity. This is included as Appendix D, and is referenced at the end of Section 4.1.
>
> > “Additionally, when the MDP is not known (RL setting), even setting the correct weights might lead to harm...”
>
> We are aware of this concern and line of work but consider it sufficiently unrelated to the core issues we study that we prefer not to add a discussion. It isn’t clear to us that this sort of learning is the key challenge in these settings when there is often ample historical data used to train models.
>
>
>
> Thank you for your thoughtful review and suggestions. We look forward to your further feedback.

---

### Review · Reviewer_ytNM · 2024-02-14

**Summary Of Contributions:**

The study proposes a reinforcement learning method for long-term group fairness; it defines a multi-objective reward and directly optimizes three objectives: maximizing decision-maker utility, maximizing group qualification, and minimizing the difference in qualification between groups. Importantly, the proposed approach aims to achieve equality without harm, i.e., avoiding reductions in qualifications for specific groups.

**Audience:**

Yes

**Broader Impact Concerns:**

No concern

**Claims And Evidence:**

Yes

**Requested Changes:**

1. Why is the qualification disparity defined as the difference between the cumulative qualification of two groups, but not the difference between instantaneous qualification, i.e., $Y_t^1-Y_t^0$? Also, many related works are aiming at reducing disparity between two groups which were not discussed in the paper, e.g., [2]

2. After defining MDP with multi-objective reward, how to find the optimal policy?

3. It is helpful to add more related works and discuss differences in more detail. For example, the authors mentioned two drawbacks that existing works suffer but only introduced one work Wen et al. (2021) that violates the "no-harm" principle.

**Strengths And Weaknesses:**

Strengths:

1. The paper proposes a framework based on reinforcement learning with a multi-objective reward function, which balances three objectives: (1) maximizing the decision-maker's utility; 2) maximizing group qualification; 3) minimizing group qualification disparity.

2. By adjusting the weights of each reward term, the authors can show theoretically that the policy doesn’t harm one or both groups (i.e., group qualification does not deteriorate).

3. The paper studies an important topic that is relevant to the audience of TMLR. It is well-written and organized.



Weaknesses:

1. While the authors claim the proposed method satisfies the "no-harm" principle, it seems that "no harm" is mostly guaranteed for one group (Thm 4.1 & 4.2), i.e., one group's qualification doesn't deteriorate. Only when natural fairness exists, can harm be avoided for both groups (Thm 4.3).

2. My major concern of the paper is its novelty. Using MDP to formulate the interactions between individuals and policy is standard and has been widely used in prior works. There are also extensive studies that simultaneously consider the welfare of both decision-maker and individuals (including qualification and group equality), e.g., [1]. To balance multiple objectives, the paper simply defines a reward function as the weighted sum of three rewards.  Since the algorithm for finding the optimal policy seems standard, I am uncertain if defining a new reward function is sufficiently novel.

3. The experiments are conducted on simulated dynamics, which may limit the impacts of the paper.

---

> ### Author Response · Authors · 2024-03-06
> **Response to feedback and corresponding changes to our revision**
>
> **Clarification on "No-Harm" Property**
>
> > "While the authors claim the proposed method satisfies the "no-harm" principle, it seems that "no harm" is mostly guaranteed for one group (Thm 4.1 & 4.2), i.e., one group's qualification doesn't deteriorate. Only when natural fairness exists, can harm be avoided for both groups (Thm 4.3)."
>
> Thank you for your insightful feedback on our discussion of the "no-harm" principle and its application within our methodology. It appears our initial description may not have fully captured the essence of our approach, especially regarding the nuanced objectives we aim to achieve with respect to fairness and qualification disparities.
>
> To clarify and directly address your concern: Our primary goal with respect to avoiding harm is to navigate the balance between improving the qualifications of one group without unjustly compromising the qualifications of another. It's crucial to underscore that in the pursuit of fairness, adjustments to the qualifications of the more advantaged group may be necessary to facilitate substantial improvements for the disadvantaged group. This is a cornerstone of the challenges with fairness, since if it were otherwise and you could improve the qualification of the disadvantaged group without consequence, achieving fairness would be trivial.
>
> The most severe problem we aim to solve is when both groups experience a reduction in qualifications in the quest for equality. Therefore, our method is designed to explore and implement solutions where a slight reduction in the qualifications of the advantaged group could lead to significant qualification gains for the disadvantaged group. Theorems 4.1 and 4.2 highlight situations where we can protect at least one group from harm, a critical step towards this goal. Theorem 4.3 clarifies that there is potential to achieve the ultimate outcome, which is no harm to either group, and that there is a specific parameterization of our approach where this is guaranteed for natural fairness.
>
> In order to make this more clear, we've revised Section 1 to clarify each of these points.
>
> **Clarification on Novelty and Methodology**
>
> > "My major concern of the paper is its novelty. Using MDP to formulate the interactions between individuals and policy is standard..also extensive studies that simultaneously consider the welfare of both decision-maker and individuals (including qualification and group equality)... To balance multiple objectives, the paper simply defines a reward function as the weighted sum of three rewards... uncertain if defining a new reward function is sufficiently novel."
>
> Our setup is intentionally similar to that of Zhang et al. As described in the related work section, their work characterizes the long-term impact of implementing localized constraints, and with respect to various environment setups (several of which are reflected in our experiment section). Essentially, Zhang et al describe the problem, but don't provide a solution to the problem. Our paper provides a solution. We've clarified this in Section 1.1 and 1.2, and in Section 5.
>
> **Definition of Qualification Disparity**
>
> > "Why is the qualification disparity defined as the difference between the cumulative qualification of two groups, but not the difference between instantaneous qualification, i.e., $Y_t^1-Y_t^0$? Also, many related works are aiming at reducing disparity between two groups which were not discussed in the paper, e.g., [2]"
>
> Our definition of qualification disparity in terms of cumulative outcomes is purposefully designed to capture the long-term effects of decision-making on group qualifications. This perspective is crucial for reflecting the broader societal and systemic shifts we aim to influence through the implementation of fairer decision-making policies. Timestep-localized qualification differences (i.e. "one-shot" constraints) fail to encapsulate the cumulative and long-term impacts of decisions, which are essential for achieving the overarching goals of fairness that guide our research. In fact, the entire line of research following Liu et al. 2018 is to show that one-shot constraints often fall short of achieving long-term fairness.
>
> Our Section 1.1-1.2 revisions should make this point more clear.
>
> **Optimal Policy Computation**
>
> > "After defining MDP with multi-objective reward, how to find the optimal policy?"
>
> To compute the optimal policy, we use linear programming. However, how the policy is computed is not of interest to us, since we design the solution by providing a reward function that, when optimized, induces the intended behavior. In other words, the whole point of our approach is to decouple the fairness objective from the RL policy-learning strategy.
>
> Thank you for your thoughtful review and suggestions. We look forward to your further feedback. Also, your review included references [1] and [2] but doesn’t appear to have the corresponding citations. We are happy to review them, however, if applicable.

---

### Review · Reviewer_HwGU · 2024-02-16

**Summary Of Contributions:**

The paper studies group fairness in reinforcement learning. The paper's stylized model is inspired by Liu et al and Zhang et al in which loan applicants from two groups arrive at each timestep and the decision maker has to make a loaning decision which in turn can change the distribution of (qualification) of applicants in the next time step. The paper aims to study the long-term effects of these decisions in particular the relationship between the decision maker's utility and the disparity between the qualification status of applicants from the two groups.

**Audience:**

Yes

**Broader Impact Concerns:**

Broader impacts is sufficiently addressed.

**Claims And Evidence:**

Yes

**Requested Changes:**

-- I would like to see more justification for the transition probabilities provided in Table 1 or at least a more detailed exploration of the parameter space.

-- Is it clear when natural fairness exists? In particular, can we quantify when the "no harm principle" can be achieved? The authors explore this empirically but can we say more?

-- Can the authors provide intuition on why the proposed solution is different compared to the previous solutions? In particular, previous work aimed at maximizing utility subject to fairness constraints. This can be written as an unconstrained problem by using Lagrangian multiplier. The current work looks at an optimization problem which is the weighted sum of utility and degree of fairness violation. To me it is unclear why this formulation can satisfy the two desiderata that the paper is after while the previous formulation cannot.

-- Intuitively why do we need two time steps? Can the analysis be done using one time step similar to Liu et al and Zhang et al.

-- Some recent work in the area such as https://openreview.net/forum?id=JkIH4MeOc3 are not cited.

**Strengths And Weaknesses:**

Strengths:

-- The paper studies a really important problem.

-- The "no harm principle" is especially important in resource allocation problems at have been largely overlooked by previous work.

Weaknesses:

-- The setting is almost identical to Zhang et al so in that sense the novelty of the formulation is limited.

-- There is no discussion of how the optimal policy can be computed. I assume this can be done in a brute-force dynamic programming style since there are only two time steps.

---

> ### Author Response · Authors · 2024-03-06
> **Response to feedback and corresponding changes to our revision**
>
> Thank you for your insightful and constructive feedback. We appreciate the opportunity to address the concerns raised and outline the improvements we've made based on your comments.
>
> **Quantifying Natural Fairness + "No-Harm Principle" & Difference from Prior Works**
>
> > "Is it clear when natural fairness exists? In particular, can we quantify when the "no harm principle" can be achieved? The authors explore this empirically but can we say more?"
>
> > "Can the authors provide intuition on why the proposed solution is different compared to the previous solutions? In particular, previous work aimed at maximizing utility subject to fairness constraints. This can be written as an unconstrained problem by using Lagrangian multiplier. The current work looks at an optimization problem which is the weighted sum of utility and degree of fairness violation. To me it is unclear why this formulation can satisfy the two desiderata that the paper is after while the previous formulation cannot."
>
> Your question about quantifying when natural fairness exists and when the "no-harm principle" can be achieved is an excellent one. Natural fairness and guarantees on harm are not known a priori. If they were, solving for fairness would be quite trivial (i.e. in the case of natural fairness, simply do nothing).
>
> Theorems 4.1-4.3 are significant, not because the decision-maker knows definitively which of them are relevant, but because the properties provide guidance based on their expectation of their situation.
>
> While we recognize that the above answer might seem vague, or half answered, it is essential to recognize that it extends beyond the scope of other works in this field. For example, Zhang et al. limit their contribution to identifying fairness properties through the application of local fairness constraints, without offering a methodology for policy learning. Our work advances this discussion by introducing a policy-learning technique via a reward function, designed to learn policies that embody these fairness properties. This represents a significant step forward in operationalizing fairness within policy learning frameworks.
>
> Also, previous methods that typically apply fairness constraints within optimization problems assume that the individual utility (i.e. qualification) is the same as the decision-maker's utility function. When you think of qualification as its own utility, which encompasses more real-world environments, this is a significant deviation from the original problem statement since there are two utility functions being optimized, alongside a constraint-based approach where the constraint is across multiple timesteps.
>
> We've made several modifications to Section 1 which should make these points more clear.
>
> **Justification for Transition Probabilities**
>
> > "I would like to see more justification for the transition probabilities provided in Table 1 or at least a more detailed exploration of the parameter space."
>
> We acknowledge the need for clearer descriptions and justification for the setup in our experiment section. These probabilities are designed to reflect a good coverage of different qualification dynamics, including those studied in contemporary papers. We've further clarified this in our revisions in Section 5.1.
>
> **Rationale for Two Time Steps**
>
> > "Intuitively why do we need two time steps? Can the analysis be done using one time step similar to Liu et al and Zhang et al."
>
> We used two timesteps to show a scenario where a decision-maker needs to consider fairness across a series of decisions, which is more reflective of the real-world scenarios we aim to address, and also more closely resembles the models that our work builds on - Zhang et al. 2020 and Mouzannar et al. 2019.
>
>
>
> **Missing citations**
>
> > "Some recent work in the area such as https://openreview.net/forum?id=JkIH4MeOc3 are not cited."
>
> We updated our Related Works section to include this.
>
> **Computing Optimal Policies**
>
> > “There is no discussion of how the optimal policy can be computed. I assume this can be done in a brute-force dynamic programming style since there are only two time steps.”
>
> As mentioned in the introduction, an advantage of our approach is that since we are only defining rewards, any standard technique can be used to find policies. For our experiments we use linear programming. See Section 5.2.
>
>
>
> Thank you for your thoughtful review and suggestions. We look forward to your further feedback.

---

### Author Response · Authors · 2024-02-22
**Discussion of primary concerns**

We thank the reviewers for their feedback. We are making revisions and will submit a revision shortly. Until then, we will discuss the main concerns here, and would appreciate feedback.

## Framing our contributions

We acknowledge we did not clearly articulate how our research builds on and differs from past work, obscuring our contributions and novelty. We clarify this here, and will update our paper accordingly.

Liu et al. (2018) highlighted how local fairness constraints might not ensure long-term fairness in decision-making. Subsequent studies by Mouzannar et al. (2019) and Zhang et al. (2020) investigated conditions when such constraints could actually promote long-term fairness, but did not offer ways to learn fair policies. Our contributions fill this gap, by proposing and characterizing a fair policy-learning strategy by embedding the objectives in a scalar reward.

Other strategies like those of Wen et al. (2021), Hu & Zhang (2022), and Chi et al. (2021) seek long-term fairness, but often lower some groups' qualifications without broader benefits, an issue termed "harm."

Martinez et al. (2020) proposed a policy-learning strategy to maximize the minimum group qualification, specifically to avoid "harm". Our method offers an alternative that yields more optimal solutions.

A reviewer compared our weighted-sum objective to similar ones in Wen et al. (2021) and Satija et al. (2023), which directly apply constraints (as opposed to the Lagrangian form), and are therefore prone to harm.

## Clarification of "harm" and our objective

Our initial description may not have fully captured the essence of our approach, especially regarding the nuanced objectives we aim to achieve with respect to "harm".

Our primary goal with respect to avoiding harm is to navigate the balance between improving the qualifications of one group without unjustly compromising the qualifications of another. Achieving fairness often requires adjusting the qualifications of the more advantaged group to aid the disadvantaged group significantly. This is the inherent challenge of fairness, since if it were always possible to improve one or more groups without consequence, achieving fairness would be trivial.

Theorems 4.1 and 4.2 highlight situations where we can avoid the harm to both groups. Theorem 4.3 clarifies that we can achieve no harm to either group (the best possible outcome) under natural fairness for a specific parameterization.

## Generalizing beyond two groups

Several reviewers noted the limitation of our approach to only two groups.

Most fairness research focuses on this setting, and we chose to concentrate on two groups because this scenario presents a foundational case that is both prevalent and critical in fairness research. Studying two groups simplifies understanding fairness and its effects, providing a foundation for exploring more complex situations.

That said, we can generalize our approach beyond $\lvert\mathcal{Z}\rvert=2$ as follows.

We keep the overall reward structure consistent with Equation (1)
$$
R(\lambda, \sigma_t) = \lambda^D R^D(\sigma_t) + \lambda^Q R^Q(\sigma_t) + \lambda^F R^F(\sigma_t) ,
$$
but we define $R^Q$ and $R^F$ as a function of the number of groups $\lvert \mathcal{Z} \rvert$:
$$
R^Q(\sigma_t) = \frac{1}{\lvert \mathcal{Z}\rvert} \sum_{i=0}^{\lvert \mathcal{Z}\rvert} Y_{t+1}^i ,
$$

$$
R^F(\sigma_t) = \frac{1}{\lvert \mathcal{Z}\rvert} \biggl(1 +\sum_{i=0}^{\lvert \mathcal{Z}\rvert} \big\lvert Y^i_{t} - Y^\texttt{avg}_{t} \big\rvert -
$$

$$
	\sum_{i=0}^{\lvert \mathcal{Z}\rvert} \big\lvert Y^i_{t+1} - Y^\texttt{avg}_{t+1} \big\rvert
\biggr)
$$

where
$$
Y_{t}^\texttt{avg} = \frac{1}{\lvert \mathcal{Z}\rvert} \sum_{i=0}^{\lvert \mathcal{Z}\rvert} Y_{t}^i  .
$$
A full theoretical analysis of this new case likely needs new ideas, as the two cases we consider in our proof become twelve cases, even with only three protected groups. However, we are working on numerical illustrations about how our approach is still relevant in this more general setting.

## Purpose of our experiments

There were several comments/concerns with our experiment section, most of which stem from the fact that we did not clearly articulate our goals, which are three-fold:

1. To visualize Theorems 4.1-4.3.
2. To illustrate how our reward encourages fairness, beyond avoiding harm.
3. To illustrate how our reward can lead to better policies than two benchmarks: one that avoids harm (Martinez et al. 2020), and a constraint-based approach that is prone to harm (Wen et al. 2021).

Our experiments, mirroring repeat loan applications with transition probabilities similar to those in Zhang et al. and Wen et al. (2021), ensure alignment with relevant literature. The core of our study is a scalar reward function suitable for any scalar RL strategy, making additional model-free RL, like deep RL, unnecessary for our goals. We plan to clarify our approach and address methodological queries from reviewers in our revision.

---

### Author Response · Authors · 2024-03-06
**Revision**

We thank each of the reviewers for their insightful and constructive feedback. We've attached our revised paper that addresses the feedback and concerns raised by the reviewers. We've provided individual responses to each of the reviewers that addresses their concerns and includes the relevant revisions we've made.

We've also included all changes made in our revision in the "Changes Since Last Submission" section above.

Thank you for your thoughtful review and suggestions. We look forward to your further feedback.

---

### Decision · Action_Editor_MKoS · 2024-03-29

**Recommendation:** Accept as is

**Comment:**

Reviewers agree the paper makes interesting contributions in an important question. The revised version addressed issues raised in initial reviews, such as clarity of contributions and experiments, and discussions of limitations. While there remain limitations, most notably the restriction to the two-group setting, the existing contributions are useful to be published.

Other comments:
* Is the condition needed in Theorem 1 that "natural fairness does not exist"? Should the conclusion still hold when natural fairness exists (Thm 3)?
* Fig 1: the diagram suggests outcomes are observed immediately at t=1. Is it a valid assumption for problems like loan decisions?

**Audience:**

The paper would be of interest to a sub-community in TMLR working on ML fairness.

**Claims And Evidence:**

The paper supports the claims through proofs and numerical experiments. The evidence is solid.